# Enhancing Logits Distillation with Plug&Play Kendall's $\tau$ Ranking Loss

Yuchen Guan [*1]   Runxi Cheng [*1]   Kang Liu [1]   Chun Yuan [1]

## Abstract

Knowledge distillation typically minimizes the Kullback–Leibler (KL) divergence between teacher and student logits. However, optimizing the KL divergence can be challenging for the student and often leads to sub-optimal solutions. We further show that gradients induced by KL divergence scale with the magnitude of the teacher logits, thereby diminishing updates on low-probability channels. This imbalance weakens the transfer of inter-class information and in turn limits the performance improvements achievable by the student. To mitigate this issue, we propose a plug-and-play auxiliary ranking loss based on Kendall's $\tau$ coefficient that can be seamlessly integrated into any logit-based distillation framework. It supplies inter-class relational information while rebalancing gradients toward low-probability channels. We demonstrate that the proposed ranking loss is largely invariant to channel scaling and optimizes an objective aligned with that of KL divergence, making it a natural complement rather than a replacement. Extensive experiments on CIFAR-100, ImageNet, and COCO datasets, as well as various CNN and ViT teacher-student architecture combinations, demonstrate that our plug-and-play ranking loss consistently boosts the performance of multiple distillation baselines. Code is available at https://github.com/OvernighTea/RankingLoss-KD

## 1. Introduction

The recent advancements in deep neural networks (DNN) have significantly enhanced performance in the field of computer vision. However, the heightened computational and

*Equal contribution [1]Tsinghua Shenzhen International Graduate School, Tsinghua University, Shenzhen, China. Correspondence to: Kang Liu <liuk22@mails.tsinghua.edu.cn>, Chun Yuan <yuanc@sz.tsinghua.edu.cn>.

*Proceedings of the 42nd International Conference on Machine Learning*, Vancouver, Canada. PMLR 267, 2025. Copyright 2025 by the author(s).

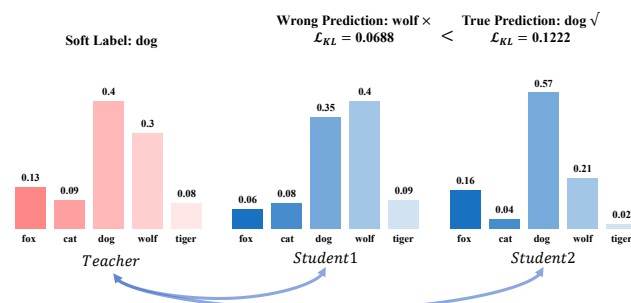

*Figure 1.* Suboptimal Case of the KL Divergence.

storage costs associated with complex networks limite their applicability. To address this, knowledge distillation (KD) has been proposed to obtain performant lightweight models. Typically, knowledge distillation involves a well-trained heavy teacher model and an untrained lightweight student model. The same data is fed to both the teacher and the student, with the teacher's outputs serving as soft labels for training the student. By constraining the student to produce predictions that match the soft labels, the knowledge in the teacher model is transferred to the lightweight student model.

Most logit-based KD methods adhere to the paradigm introduced by Hinton (Hinton et al., 2015), employing the Kullback-Leibler (KL) divergence (Kullback & Leibler, 1951) to align the logits output by the student and teacher models:

$$\mathcal{L}_{KL}(q^t \parallel q^s) = \sum_{i=1}^{C} q_i^t \cdot log(\frac{q_i^t}{q_i^s}). \qquad (1)$$

Where $q_t, q_s \in \mathbb{R}^{1 \times C}$ is the prediction vectors of teacher and student. The optimization goal of KL divergence is to achieve identical outputs between the student and teacher, thereby transferring as much knowledge as possible from the teacher to the student. However, the optimization process of KL divergence is not easy, as it is prone to suboptimal points, which can hinder further improvement in student performance. As illustrated in Fig. 1, compared to Student 2, Student 1 exhibits a smaller KL divergence with the teacher; however, Student 2 achieves the correct classification result

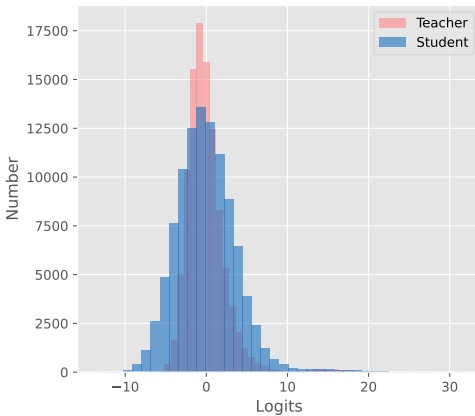

Figure 2. Most Channels in the Distribution are Low-Probability Channels.

consistent with the teacher, while Student 1 does not. This indicates that the optimization direction of KL divergence sometimes diverges from the task objective, leading students to suboptimal points.

Intuitively, and as a preliminary observation, Eq. 1 shows that in the KL divergence each channel is weighted by the predicted probability of the teacher. Channels with small $q_i^t$ therefore contribute marginally to the loss, causing the student to tend to largely ignore them (the formal derivation is provided in Appendix A.5). In practice, most channels in a single prediction have lower probabilities, as illustrated in Fig. 2. Neglecting these channels hampers knowledge transfer for two reasons:

**R1:** Gradient attenuation. Because low-probability channels carry tiny weights, the corresponding gradients are strongly damped, which makes alignment with the teacher network difficult..
**R2:** Impaired capture of inter-class information. Low-probability channels retain rich cross-category correlation information in the teacher logits; overlooking them deprives the student of these semantic correlations. For example, in Fig. 1, Student 1 may not learn the distinction between the cat and fox classes.

Considering these challenges, we aim to find an auxiliary method that mitigates the issues caused by channel scale differences, learns inter-class relationship information, and avoids the suboptimal points that KL divergence might encounter while maintaining the optimization objective of KL divergence. To address this problem, we propose a ranking-based plug-and-play auxiliary loss. The benefits of imposing ranking constraints are as follows:

**B1:** Gradients from the ranking term depend only weakly on the absolute channel scale.
**B2:** Ordered channel scores make cross-category relationships explicit.

**B3:** Fixing relative channel order helps the model escape suboptimal solutions.

Therefore, we propose to constrain the channel ranking similarity between student and teacher. We construct a plug-and-play ranking loss function based on Kendall's $\tau$ Coefficient. This ranking loss can supplement the attention to low-probability channels in the logits and provide inter-class relationship information. We demonstrate that the gradients provided by KL divergence are related to channel scale, whereas the proposed ranking loss is not. We also show that the optimization objective of the proposed ranking loss is consistent with that of KL divergence, and visualize the loss landscape to illustrate that the ranking loss helps avoid suboptimal points in the early stage and does not alter the optimization objective in the end. Extensive experiments on CNN and ViT show that the proposed plug-and-play ranking loss can enhance the performance of various logit-based methods. In conclusion, our contributions are as follows:

- We introduce a plug-and-play ranking loss for assisting knowledge distillation tasks, addressing the issues of KL divergence's neglect of low-probability channels and its tendency to fall into suboptimal points, while also learning inter-class relationship information to further enhance student performance.

- We demonstrate that the gradients provided by KL divergence exhibit sensitivity to channel scales, while the proposed ranking loss remains more robust to such variations. We also prove and visualize that the optimization objective of the proposed ranking loss is consistent with that of KL divergence and helps avoid the suboptimal points of KL divergence.

- Extensive experiments are conducted on a variety of CNN and ViT teacher-student architectures using the CIFAR-100, ImageNet, and MS-COCO datasets. Our findings confirm the widespread effectiveness of the proposed ranking loss in various distillation tasks, and its role as a plug-and-play auxiliary function provides substantial support for the training of distillation tasks.

## 2. Related Works

**Knowledge Distillation.** Knowledge distillation, initially proposed by Hinton (Hinton et al., 2015), serves as a method for model compression and acceleration, aiming to transfer knowledge from a heavy teacher model to a lightweight student model. By feeding the same samples, the teacher can produce soft labels, and training the student with these soft labels allows the transfer of knowledge from the teacher model to the student. Knowledge distillation tasks can be mainly divided into two categories: feature-based distillation (Chen et al., 2022; Zhang & Ma, 2021; Yang et al.,

2022c; Guo et al., 2023; Park et al., 2019; Yang et al., 2022b) and logit-based distillation (Hinton et al., 2015; Li et al., 2023; Zhao et al., 2022; Jin et al., 2023; Sun et al., 2024; Chi et al., 2023; Wen et al., 2021; Yang et al., 2023b). Feature-based distillation additionally utilizes the model's intermediate features, providing more information to the student and enabling the student to learn at the feature level from the teacher, often resulting in better performance. Considering safety and privacy, the intermediate outputs of the model are often not obtainable; hence, logit-based methods that only use the model outputs for distillation offer better versatility and robustness. Our proposed method can act as a plug-and-play module added to logit-based methods, offering higher flexibility and further enhancing the performance of logit-based methods.

Recent knowledge distillation methods have found that overly strict constraints can sometimes hinder the student's transfer of knowledge from the teacher's soft labels. For instance, (Cao et al., 2022) discovered that differences in feature sizes in feature-based methods could limit the student's learning; while (Sun et al., 2024) found that using the same temperature for both teacher and student in logit-based methods could affect further improvements in student performance. To reduce the learning difficulty for the student, (Sun et al., 2024; Cao et al., 2022) provided methods to ease the logit matching difficulty between student and teacher, yet we find that methods providing additional guidance to the student have not been fully explored.

**Ranking Loss in Knowledge Distillation.** For knowledge distillation, ranking loss is a relaxed constraint that can provide rich inter-class information. It was first applied in the distillation of recommendation systems (Reddi et al., 2021; Tang & Wang, 2018; Choi et al., 2021; Qin et al., 2023; Yang et al., 2022a), (Li et al., 2022) explored the application of ranking loss in object detection tasks, and (Gao et al., 2020) discussed the role of ranking in the distillation of language model tasks. However, the exploration of ranking loss in logit-based image classification task distillation is not yet comprehensive. We find that the KL divergence used for distillation in classification tasks tends to overlook information from low-probability channels and may lead to suboptimal results. Our proposed method leverages ranking loss to balance the model's attention between high-probability and low-probability channels, while also using inter-class relationships to help the model avoid suboptimal outcomes.

## 3. Preliminary

Most logit-based distillation methods adhere to the original KD proposed by Hinton (Hinton et al., 2015), which transfers knowledge by matching the logit outputs of the student and teacher. This setting is more generalizable,

allowing for distillation solely through outputs when the internal structures of the student and teacher are invisible. For a given dataset $\mathcal{D}$, assuming there are $C$ categories and $N$ samples, we possess a teacher model $f_t$ and a student model $f_s$. For a given sample $\mathcal{I} \in \mathcal{D}$, we can obtain the outputs of the teacher and student model, denoted as $z^t = f_t(\mathcal{I}), z^s = f_s(\mathcal{I})$ where $z^t, z^s \in \mathbb{R}^{1 \times C}$. Through a softmax function with temperature, the outputs are processed into the prediction vectors $q_t, q_s \in \mathbb{R}^{1 \times C}$ finally:

$$q_i^t = \frac{exp(z_i^t/\mathcal{T})}{\sum_{j=1}^{C} exp(z_j^t/\mathcal{T})}, \quad q_i^t = \frac{exp(z_i^s/\mathcal{T})}{\sum_{j=1}^{C} exp(z_j^s/\mathcal{T})} \quad (2)$$

where $\mathcal{T}$ is a temperature parameter, $z_i$ represents the logit value of the $i$-th channel of the model output, $q_i$ represents the predicted probability for the target being the $i$-th class. The KL divergence loss function used to constrain the student and teacher logits is of the following form:

$$\mathcal{L}_{KL}(q^t \| q^s) = \sum_{i=1}^{C} q_i^t \cdot log(\frac{q_i^t}{q_i^s}) \quad (3)$$

It can be observed that in Eq. 3, the importance of the match between the student and teacher at the $i$-th channel is influenced by the coefficient $q_i^t$. This implies that channels with smaller logit values receive less attention, leading to the KL divergence's disregard for smaller channels.

## 4. Ranking Loss Based on Kendall's $\tau$ Coefficient

In this section, we introduce the plug-and-play ranking loss function designed to constrain the ranking of channels in the logits output by the student model. With the aid of the ranking loss, the knowledge distillation task can balance the overemphasis on larger logit values and the neglect of smaller ones as measured by the KL divergence. Additionally, the ranking loss imposes a constraint on the leading channel value, which helps to correct the optimization direction and avoid suboptimal solutions. In Sec. 4.1, we will introduce the ranking loss function and employ Kendall's $\tau$ coefficient to compute the ordinal consistency between the teacher and student logits. In Sec. 4.2, we discuss how ranking loss benefits optimizing logits distillation with KL divergence. Furthermore, we discuss the differentiable form of Kendall's $\tau$ coefficient and, based on this, design three distinct forms of ranking loss functions in Appendix A.3.

### 4.1. Differentiable Kendall's $\tau$ Coefficient

In order to make the loss function pay more attention to the information provided by low-probability channels as well as to help correct the suboptimal problem in Fig. 1, we introduce the ranking loss based on Kendall's $\tau$ coefficient.

By pairing each of the $C$ channels, we can obtain $\frac{C(C-1)}{2}$ pairs, whose Kendall's $\tau$ coefficient is expressed as follows:

$$\tau = \frac{P_c - P_d}{\frac{1}{2}C(C-1)} \quad (4)$$

where $P_c$ represents concordant pairs and $P_d$ represents discordant pairs. Kendall's $\tau$ coefficient provides an expression of ordinal similarity. For the teacher and student logits, a channel pair $(i, j)$ is considered concordant if the signs of $(z_i^t - z_j^t)$ for the teacher and $(z_i^s - z_j^s)$ for the student are the same; otherwise, the pair is discordant. We substitute the logits of the teacher and student in the following manner:

$$\tau = \frac{\sum_i \sum_{j<i} sgn(z_i^t - z_j^t) \cdot sgn(z_i^s - z_j^s)}{\frac{1}{2}C(C-1)} \quad (5)$$

where $sgn()$ represents sign function. While we have quantified the ordinal similarity between the teacher and student logits, the aforementioned formula is non-smooth. To utilize the ordinal similarity for gradient computation, we approximate the sign function with the $tanh$ function to convert it into a differentiable form:

$$\tau_d = \frac{\sum_i \sum_{j<i} tanh(k \cdot (z_i^t - z_j^t)) \cdot tanh(k \cdot (z_i^s - z_j^s))}{\frac{1}{2}C(C-1)}$$

$$= \frac{2}{C(C-1)} \cdot \sum_i \sum_{j<i} \left(1 - \frac{2}{1 + e^{2 \cdot (z_i^t - z_j^t) \cdot k}}\right) \cdot$$

$$\left(1 - \frac{2}{1 + e^{2 \cdot (z_i^s - z_j^s) \cdot k}}\right) \quad (6)$$

where $k$ is a parameter that controls the steepness of the function; a larger $k$ causes the $tanh$ function to more closely approximate the sign function. By negating the similarity measure in Eq. 6, it can serve as a loss function (bounded within $[-1, 1]$) to enforce the consistency of the logits order between the teacher and student:

$$\mathcal{L}_{RK} = -\tau_d \quad (7)$$

The overall loss function is formulated as follows:

$$\mathcal{L} = \alpha \mathcal{L}_{KL} + \beta \mathcal{L}_{CE} + \gamma \mathcal{L}_{RK} \quad (8)$$

where $\mathcal{L}_{KL}$ denotes KL divergence loss, $\mathcal{L}_{CE}$ denotes Cross-Entropy loss, $\mathcal{L}_{RK}$ denotes our ranking loss, and $\alpha, \beta, \gamma$ are hyper-parameters.

## 4.2. How Ranking Loss Benefits Optimizing Logits Distillation with KL Divergence

### 4.2.1. FROM THE PERSPECTIVE OF GRADIENT: RANKING LOSS CARES ABOUT LOW-PROBABILITY CHANNELS

In this section, we analyze why ranking loss cares about low-probability channels. The gradient of KL divergence and

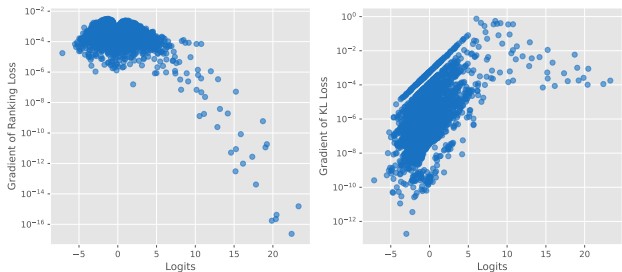

*Figure 3.* The Gradient of Ranking Loss and KL Divergence for Different Values of Logits.

ranking loss is written as follows. The specific calculation process of the gradient can be found in the appendix A.5:

$$\frac{\partial \mathcal{L}_{\text{KD}}}{\partial z_i^s} = -T\left(q_i^t - q_i^s\right). \quad (9)$$

$$\frac{\partial \mathcal{L}_{RK}}{\partial z_i^s} = -\frac{k}{C(C-1)} \sum_{j \neq i} \left[1 - \tanh^2\left(k(z_i^s - z_j^s)\right)\right] \cdot$$

$$\tanh\left(k(z_i^t - z_j^t)\right) \quad (10)$$

The gradient scale of the KL divergence is influenced by the scales of the teacher and student models. Consequently, for outputs from low-probability channels, when the scales of the teacher and student are similar, the KL divergence's gradient for these components becomes negligible, leading to the neglect of information from low-probability channels. Although temperature scaling is typically employed to address this issue, it uniformly amplifies the gradients of all logits, thereby continuing to overlook the outputs of low-probability channels. In contrast, the gradients produced by ranking loss will be less affected by the scale of the teacher's logits. The gradient of a logit channel primarily depends on the difference between its rank and the target rank, effectively harnessing the knowledge from low-probability channels. Therefore, In the early stage of training, ranking loss helps the model quickly learn the overall ranking of the teacher model, which helps the model converge to a better initial solution faster.

In our study, we conduct a visualization of the gradients associated with both the ranking loss and the KL divergence loss in Fig .3. The logits of the student model are randomly generated but maintain the same scale as those of the teacher model across each channel. Our findings indicate that the ranking loss consistently provides gradients of similar scale across varying logit values, whereas the gradients from the KL loss are heavily dependent on the specific logit values. This suggests that the ranking loss offers a more uniform attention distribution across different logits compared to the KL loss. Notably, the ranking loss assigns low-probability gradients to logits with larger values, as they are the clas-

*Table 1.* CIFAR-100 Heterogeneous Architecture Results. The Top-1 Accuracy (%) is reported as the evaluation metric. The teacher and student have heterogeneous architectures. We incorporate the proposed ranking loss into the existing logit-based methods, with the performance gains indicated in parentheses. The best and second best results are emphasized in **bold** and underlined.

| KD | Teacher
Student | ResNet32×4
79.42
WRN-16-2
73.26 | ResNet32×4
79.42
WRN-40-2
75.61 | ResNet50
79.34
MN-V2
64.60 | ResNet32×4
79.42
SHN-V1
70.50 | WRN-40-2
75.61
SHN-V1
70.50 |
|---|---|---|---|---|---|---|
| | FitNet (Adriana et al., 2015) | 74.70 | 77.69 | 63.16 | 73.59 | 73.73 |
| | CRD (Tian et al., 2019) | 75.65 | 78.15 | 69.11 | 75.11 | 76.05 |
| | ReviewKD (Chen et al., 2021) | 76.11 | 78.96 | 69.89 | 77.45 | 77.14 |
| | DIST (Huang et al., 2022) | 75.58 | 78.02 | 68.66 | 76.34 | 76.00 |
| | LSKD (Sun et al., 2024) | **77.53** | 79.66 | 71.19 | 76.48 | 76.93 |
| | KD (Hinton et al., 2015) | 74.9 | 77.7 | 67.35 | 74.07 | 74.83 |
| | KD+Ours | 75.18(+0.28) | 78.50(+0.80) | 70.45(+3.10) | 75.98(+1.91) | 76.13(+1.30) |
| | CTKD (Li et al., 2023) | 74.57 | 77.66 | 68.67 | 74.48 | 75.61 |
| | CTKD+Ours | 75.71(+1.14) | 78.61(+0.95) | 70.18(+1.51) | 76.67(+2.19) | 76.80(+1.19) |
| | DKD (Zhao et al., 2022) | 75.7 | 78.46 | 70.35 | 76.35 | 76.33 |
| | DKD+Ours | 75.99(+0.29) | 78.75(+0.29) | 70.90(+0.55) | 77.36(+1.01) | 76.43(+0.10) |
| | MLKD (Jin et al., 2023) | 76.52 | 79.26 | 71.04 | 77.18 | 77.44 |
| | MLKD+Ours | 76.83(+0.31) | **79.86(+0.60)** | **71.66(+0.62)** | **77.63(+0.45)** | **77.87(+0.43)** |

sification targets and reach the correct rank. Consequently, these logits receive less attention from the ranking loss.

### 4.2.2. FROM THE PERSPECTIVE OF OPTIMAL SOLUTION DOMAIN: RANKING LOSS WON'T INTERFERE WITH KL DIVERGENCE AT THE END

In this section, we analyze the optimal solution domain for Kullback-Leibler (KL) divergence and ranking loss to show how ranking loss affects the optimization process of KL divergence. For KL divergence, the optimal solution domain is achieved when the distribution of the student model aligns perfectly with that of the teacher model, which is equivalent to a linear mapping on the logits:

$$q_i^t = q_i^s \quad \forall i \quad \Rightarrow \quad z_i^t = z_i^s + c \quad \forall i \qquad (11)$$

$q_i^t$ and $q_i^s$ respectively represent the probability outputs of the teacher and the student for the i-th class, and $c$ represents any constant number. For the ranking loss, the optimal solution domain is such that for any two indices, the order of the logits output by the teacher and the student is consistent:

$$sgn(z_i^t - z_j^t) \cdot sgn(z_i^s - z_j^s) = 1 \quad \forall i,j$$
$$\iff (z_i^t - z_j^t) \cdot (z_i^s - z_j^s) > 0 \quad \forall i,j \qquad (12)$$
$$\iff z_i^t = F(z_i^s) \quad \forall i \quad where \quad F'(x) > 0$$

It implies that the optimal solution domain for ranking loss is quite lenient and easily attainable, encompassing the optimal solution domain for KL divergence. Therefore, in the later stage of training, the ranking loss will not hinder the optimization of KL divergence, which means our ranking loss can perfectly be used as an auxiliary loss.

### 4.2.3. FROM THE PERSPECTIVE OF CLASSIFICATION: RANKING LOSS HELPS STUDENT CLASSIFY CORRECTLY

As illustrated in Fig. 1, using KL loss may lead to a smaller loss yet result in incorrect classification. The tendency to fall into local optima will hinder the optimization process in distillation. Although the use of cross-entropy loss can mitigate this issue, it does not incorporate information from the teacher model. Consequently, in experiments, cross-entropy loss is often assigned a very small weight, which also leads to a small gradient. For instance, KD (Hinton et al., 2015)$\mathcal{L}_{CE} : \mathcal{L}_{KL} = 0.1 : 0.9$; DKD (Zhao et al., 2022) $\mathcal{L}_{CE} : \mathcal{L}_{TCKL} : \mathcal{L}_{NCKL} = 1 : 1 : 8$; MLKD (Jin et al., 2023) $\mathcal{L}_{CE} : \mathcal{L}_{KL} = 0.1 : 9$; LSKD (Sun et al., 2024) $\mathcal{L}_{CE} : \mathcal{L}_{KL} = 0.1 : 9$. As a result, cases with lower KL loss but incorrect classification still frequently occur. In contrast, by aligning the rank between student and teacher, ranking loss helps KL divergence avoid such suboptimal situations. Also, the ranking loss can incorporate information from the teacher model. In this way, ranking loss helps students classify correctly, avoiding the suboptimal case in logit distillation. Furthermore, a model with better generalization should have a more reasonable rank of logits. For instance, recognizing that a tiger is more similar to a cat than to a fish. By learning such inter-class relationships, the student can improve classification performance and enhance its representational capacity.

## 5. Experiments

**Datasets.** In order to validate the efficacy and robustness of our proposed method, we conduct extensive experiments

*Table 2.* CIFAR-100 Homogenous Architecture Results. The Top-1 Accuracy (%) is reported as the evaluation metric. The teacher and student have homogeneous architectures. We incorporate the proposed ranking loss into the existing logit-based methods, with the performance gains indicated in parentheses. The best and second best results are emphasized in **bold** and underlined.

| KD | | ResNet32×4 | VGG13 | WRN-40-2 | ResNet110 |
|---|---|---|---|---|---|
| | Teacher | 79.42 | 74.64 | 75.61 | 74.31 |
| | Student | ResNet8×4 | VGG8 | WRN-40-1 | ResNet20 |
| | | 72.50 | 70.36 | 71.98 | 69.06 |
| | FitNet (Adriana et al., 2015) | 73.50 | 71.02 | 72.24 | 68.99 |
| | CRD (Tian et al., 2019) | 75.51 | 73.94 | 74.14 | 71.46 |
| | ReviewKD (Chen et al., 2021) | 75.63 | 74.84 | 75.09 | 71.34 |
| | DIST (Huang et al., 2022) | 76.31 | 73.80 | 74.73 | 71.40 |
| | LSKD (Sun et al., 2024) | **78.28** | 75.22 | 75.56 | 72.27 |
| | KD (Hinton et al., 2015) | 73.33 | 72.98 | 73.54 | 70.67 |
| | KD+Ours | 74.74(+1.41) | 74.14(+1.16) | 74.49(+0.95) | 71.09(+0.42) |
| | CTKD (Li et al., 2023) | 73.39 | 73.52 | 73.93 | 70.99 |
| | CTKD+Ours | 75.59(+2.2) | 74.76(+1.24) | 74.86(+0.93) | 71.08(+0.09) |
| | DKD (Zhao et al., 2022) | 76.32 | 74.68 | 74.81 | 71.06 |
| | DKD+Ours | 76.61(+0.29) | 75.1(+0.42) | 74.94 (+0.13) | 71.84(+0.78) |
| | MLKD (Jin et al., 2023) | 77.08 | 75.18 | 75.35 | 71.89 |
| | MLKD+Ours | 77.25(+0.17) | **75.35(+0.17)** | **76.08(+0.73)** | **72.35(+0.46)** |

on three datasets. 1) CIFAR-100 (Krizhevsky et al., 2009) is a significant dataset for image classification, comprising 100 categories, with 50,000 training images and 10,000 test images. 2) ImageNet (Russakovsky et al., 2015) is a large-scale dataset utilized for image classification, comprising 1,000 categories, with approximately 1.28 million training images and 50,000 test images. 3) MS-COCO (Lin et al., 2014) is a mainstream dataset for object detection comprising 80 categories, with 118,000 training images and 5,000 test images.

**Baselines.** As a plug-and-play loss, we apply the proposed ranking loss to various logit-based methods, including KD (Hinton et al., 2015), CTKD (Li et al., 2023), DKD (Zhao et al., 2022), and MLKD (Jin et al., 2023), to verify whether it can bring performance gains to knowledge distillation methods. We also compare it with various feature-based methods, including FitNet (Adriana et al., 2015), CRD (Tian et al., 2019), and ReviewKD (Chen et al., 2021). Additionally, we compare it with other auxiliary losses and modified KL divergence methods, including DIST (Huang et al., 2022) and LSKD (Sun et al., 2024).

**Implementation Details.** To ensure the robustness of the proposed plug-and-play loss without introducing excessive configurations, we maintain the same experimental settings as the baselines used (KD+Ours and KD share the same experimental setups for example). We set the batch size to 64 for CIFAR-100, 512 for ImageNet and 8 for COCO. We employ SGD (Sutskever et al., 2013) as the optimizer, with the number of epochs and learning rate settings consistent with the comparative baselines. The hyper-parameters $\alpha$, $\beta$

in Eq. 6 are set to be the same as the compared baselines to maintain fairness, and $\gamma$ are set equal to $\alpha$. We utilize 1 NVIDIA GeForce RTX 4090 to train models on CIFAR-100 and 4 NVIDIA GeForce RTX 4090 for training on ImageNet. The algorithm of our method can be found in Appendix A.4.

**Other Settings.** In order to prevent the occasional gradient explosions at the start of optimization due to the too variant logits caused by random initialization of student, we add and **only** add normalization in ranking loss, which does not affect our fairness as a plug-and-play function. We conduct experiments about ablation of normalization in Appendix A.2, and the algorithm can be found in Appendix A.4.

### 5.1. Main Result

**CIFAR-100 Results.** In our study, we conduct a comparative analysis of Knowledge Distillation (KD) outcomes across various teacher-student (He et al., 2016; Simonyan & Zisserman, 2014; Zagoruyko & Komodakis, 2016; Zhang et al., 2018; Howard et al., 2017; Sandler et al., 2018) configurations. While Tab. 1 presents cases where the teacher and student models share the heterogeneous architecture, Tab. 2 illustrates instances of homogenous structures. Furthermore, as a plug-and-play method, we applied ranking loss in multiple logit-based distillation techniques. The incorporation of ranking loss resulted in an average improvement of 1.83% in vanilla KD. In the context of existing state-of-the-art (SOTA) logits-based methods, including DKD, CTKD, and MLKD, significant gains are made.

**ImageNet Results.** The results on ImageNet of KD in terms of top-1 and top-5 accuracy are compared in Tab. 3.

*Table 3.* The top-1 and top-5 accuracy (%) on the ImageNet validation set. The teacher and student are ResNet50 and MN-V1

|  | AT | OFD | CRD | KD | KD+Ours |
|---|---|---|---|---|---|
| Top-1 | 69.56 | 71.25 | 71.37 | 70.50 | **71.54(+1.04)** |
| Top-5 | 89.33 | 90.34 | 90.41 | 89.80 | **90.84(+1.04)** |

*Table 4.* Knowledge Distillation for Transformer. The Top-1 Accuracy (%) on the validation set of CIFAR-100. The Teacher model is ResNet56.

| Student | DeiT-Tiny 65.08 | T2T-ViT-7 69.37 | PiT-Tiny 73.58 | PVT-Tiny 69.22 |
|---|---|---|---|---|
| KD | 71.11 | 71.72 | 74.03 | 72.46 |
| KD+Ours | **73.25(+2.14)** | **72.49(+0.77)** | **74.33(+0.30)** | **73.42(+0.96)** |

Our proposed method can achieve consistent improvement on the large-scale dataset as well. Additional ImageNet experiments are provided in Appendix A.9

### 5.2. Extensions

**Knowledge Distillation for Transformer.** To validate the effectiveness of our plug-and-play ranking loss on ViT and to assess its performance when facing larger teacher-student structural differences, we conduct experiments using ViT students. The experimental results indicate that the incorporation of ranking loss yields significant improvements over the conventional knowledge distillation, as shown in Tab. 4. This denotes that our proposed method is also applicable to distillation challenges predicated on transformer architectures. The implementation details of the Transformer experiments are attached in the Appendix.

**Knowledge Distillation for Object Detection.** To further verify the generality of our proposed method in downstream tasks, we also conduct experiments under the setting of object detection. The results show that our proposed ranking loss can improve the performance of knowledge distillation method in object detection and achieve better performance than the same period of the feature-based object detection method, which is shown in Tab. 5.

**Ablation of Weight and Steepness Parameter.** We conduct extensive ablation studies to investigate the effectiveness of ranking loss under various settings of $k$ and coefficients. Tab. 7 presents the distillation outcomes across different $k$ configurations. It is observed that a larger $k$ value yields superior performance, suggesting that the differential form of ranking loss more closely approximates the Kendall $\tau$ coefficient, thereby imparting stronger ranking knowledge. Tab. 6 delineates the performance of ranking loss under varying coefficients. This setting effectively aligns the classification outcomes of the teacher and student models, thus significantly enhancing the distillation effect.

*Table 5.* Knowledge Distillation for Object Detection. All experiments are conducted on COCO2017 with the teacher as ResNet50 and the student as MobileNet-V2.

| Method | AP | AP50 | AP75 |
|---|---|---|---|
| KD (Hinton et al., 2015) | 30.13 | 50.28 | 31.35 |
| FitNet (Adriana et al., 2015) | 30.20 | 49.80 | 31.69 |
| FGFI (Wang et al., 2019) | 31.16 | 50.68 | 32.92 |
| CTKD (Li et al., 2023) | 31.39 | 52.34 | 31.35 |
| KD+Ours | **31.99(+1.86)** | **53.80(+3.52)** | **33.37(+2.02)** |

**Ablation of Temperature.** There has been extensive and detailed research on the temperature of the KL loss (e.g., CTKD(Li et al., 2023), MLKD(Jin et al., 2023), LSKD(Sun et al., 2024)), achieving significant results. Unlike these studies, our approach aims to move away from focusing on the KL divergence and instead explore plug-and-play auxiliary losses to guide the KL divergence. Therefore, in our experiments, the temperature and other parameters of the KL divergence are kept the same as those in the respective baselines (with dynamic/multi-level temperatures used in CTKD+Ours and MLKD+Ours). The proposed ranking loss does not have a temperature parameter because the temperature does not affect the order of logits. Instead, the control over the distribution can be achieved by manipulating the steepness parameter $k$ of the sign function, and ablation experiments for $k$ are shown in Tab. 7. Meanwhile, although the temperature helps control the logit distribution, additional measures are needed to prevent the KL divergence from falling into suboptimal. We conducted an ablation study on the temperature, as shown in Tab. 8. The results demonstrate that ranking loss as a plug-and-play loss can bring further improvements under multiple temperature settings.

**Comparison between Ranking loss and KL Divergence.** We separately evaluate distillation with KL divergence alone and with the proposed ranking loss alone; the results are presented in Tab. 9. We observe that using the ranking loss alone achieves performance comparable to using KL divergence alone, and that combining the two objectives yields further gains. These findings substantiate the efficacy of the proposed ranking loss.

**More Experiments.** Additional experiments and discus-

*Table 6.* Ablation of Weight. The Top-1 Accuracy (%) on the validation set of CIFAR-100.

| Teacher | ResNet32×4 79.42 | WRN-40-2 75.61 | ResNet32×4 79.42 | ResNet50 79.34 |
|---|---|---|---|---|
| Student | ResNet8×4 72.50 | WRN-40-1 71.98 | SHN-V2 71.28 | MN-V2 64.60 |
| KD(Baseline) | 73.33 | 73.54 | 74.45 | 67.35 |
| $\gamma = 0.1$ | 74.15 | 74.15 | 75.52 | 69.25 |
| $\gamma = 0.5$ | 74.84 | 74.07 | 76.34 | 69.81 |
| $\gamma = 0.9$ | 74.74 | 74.49 | 76.58 | 70.45 |
| $\gamma = 2$ | 74.62 | **74.65** | **77.07** | 69.97 |
| $\gamma = 4$ | **75.07** | 73.60 | 77.05 | **70.59** |
| $\gamma = 6$ | 74.78 | 73.09 | 76.90 | 69.58 |

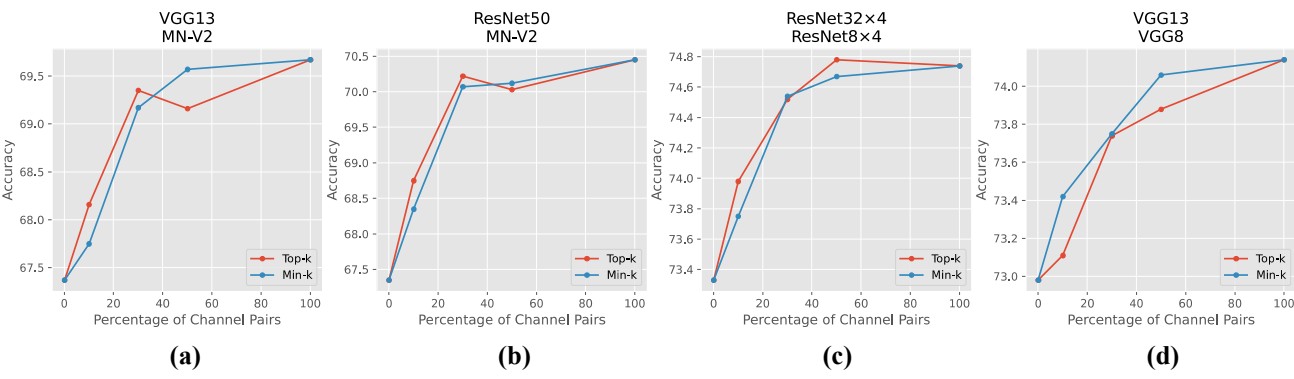

*Figure 4.* Ranking Loss using Top-K and Min-K channels. Top-1 Accuracy (%) of Top-k/Min-k Ranking Loss. 0 and 100 on the x-axis represent KD and KD+Rank, respectively.

*Table 7.* Ablation of $k$. The Top-1 Accuracy (%) on the validation set of CIFAR-100.

| | ResNet32×4 | WRN-40-2 | ResNet32×4 | ResNet50 |
|---|---|---|---|---|
| Teacher | 79.42 | 75.61 | 79.42 | 79.34 |
| | ResNet8×4 | WRN-40-1 | SHN-V2 | MN-V2 |
| Student | 72.50 | 71.98 | 71.28 | 64.60 |
| KD(Baseline) | 73.33 | 73.54 | 74.45 | 67.35 |
| $k=0.1$ | 73.13 | 73.75 | 75.47 | 68.9 |
| $k=0.5$ | 74.36 | 74.17 | 75.73 | 68.87 |
| $k=1$ | 74.74 | 74.49 | 76.58 | 70.45 |
| $k=2$ | 74.79 | **74.87** | 77.06 | 70.53 |
| $k=4$ | 75.56 | 74.48 | **77.21** | **70.81** |
| $k=6$ | **75.74** | 74.11 | 76.11 | 70.32 |

*Table 8.* Ablation of Temperature. All experiments are conducted on CIFAR-100 with the teacher as ResNet32×4 and the student as ResNet8×4.

| Temperature | T = 4 | T = 5 | T = 6 | T = 10 |
|---|---|---|---|---|
| KD | 73.33 | 73.39 | 73.43 | 73.55 |
| KD+Ours | 73.56(+0.23) | 73.49(+0.10) | 74.36(+0.93) | 74.04(+0.49) |

sions, including *Derivation of the KL Loss and Ranking Loss*, *Ablation of Normalization*, and *Different Forms of Ranking Loss*, are provided in the Appendix. Please refer to the Sec. A for further details.

### 5.3. Analysis

**Accuracy & Loss Curves with Ranking Loss.** The accuracy and loss curves of KD and KD+Ours, as shown in Fig. 5, demonstrate how ranking loss aids in optimization. The middle figure shows that the precise alignment of KL divergence also makes channel ranking more ordered, but adding ranking loss achieves a more consistent ranking more quickly. The right figure shows that in the early stages, ranking loss accelerates the reduction of KL loss and reduces its oscillation in suboptimal regions. In the later stages, ranking loss does not interfere with KL divergence and ultimately reaches a better position. The left figure shows that with ranking loss, student achieves leading accuracy at all stages. This indicates that the addition of the ranking loss helps

*Table 9.* Experiments on Combining DIST.

| Teacher | ResNet32×4 | ResNet32×4 | WRN-40-2 | VGG13 | WRN-40-2 |
|---|---|---|---|---|---|
| Student | SHN-V1 | WRN-40-2 | SHN-V1 | VGG8 | WRN-40-1 |
| KD (Only KL) | 74.07 | 77.70 | 74.83 | 72.98 | 73.54 |
| Only Ranking Loss | 75.60 | 78.04 | 75.81 | 73.72 | 74.00 |
| KL+Ranking Loss | 75.98 | 78.50 | 76.13 | 74.14 | 74.49 |

the model converge and achieve better generalization and performance.

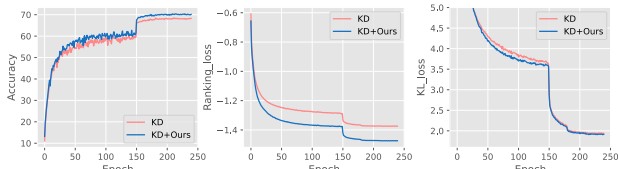

*Figure 5.* Accuracy & Loss Curves with Ranking Loss. **Left:** Top-1 Test Accuracy (%) Curve. **Middle:** Loss Curve of Ranking Loss. **Right:** Loss Curve of KL Divergence.

**Visualization of Loss Landscape.** To further investigate the role of ranking loss in the distillation process, we visualized the loss landscapes (Li et al., 2018) of student models with and without the application of ranking loss during Knowledge Distillation (KD), as depicted in Fig. 4. It is evident that the student models distilled with ranking loss exhibit a markedly flatter loss landscape and fewer local optima compared to those without it. We hypothesize that during the optimization process, ranking loss can filter out certain local optima that, despite presenting a better overall loss performance, yield poorer classification outcomes. Consequently, ranking loss can effectively enhance the generalization performance of student models.

**Top-k & Min-k Ranking.** To further validate the beneficial knowledge present in low-probability channels, we conducted comparative experiments using the top 10%, top 30%, top 50%, and min 10%, min 30%, min 50% channels. The experiments were performed across four combinations of homogeneous and heterogeneous teacher-student pairs, and the results are presented in Fig. 4. We observed that

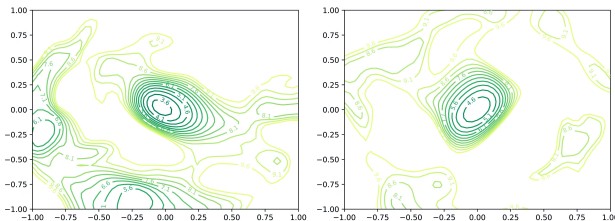

*Figure 6.* Loss Landscape. **Left:** The left landscape shows the suboptimal solutions in the distillation task. **Right:** After adding our ranking loss, the suboptimal solution is significantly reduced, as shown on the right.

using the min-k channels achieves results similar to those obtained with top-k channels (as shown in Fig. 4(a),(b), and (c)). Additionally, in some cases, min-k channels provide even more beneficial information to aid student learning (as demonstrated in Fig. 4(d)), which indicates that the low-probability channels also contain rich knowledge.

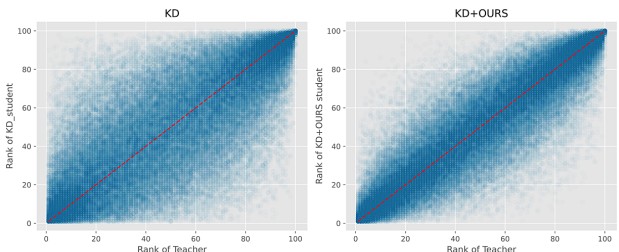

*Figure 7.* Ranking Alignment Comparison. **Left:** KD. **Right:** KD+Ours.

**Further visualization of Ranking.** We show in Fig. 3 that ranking loss balances the focus on high-probability and low-probability channels. To further illustrate the role of ranking loss, we present a visualization of the ranking results generated by KD and KD+Ours in Fig. 7. It can be seen that after adding ranking loss, the channels are more ordered overall, which is more obvious in the low-probability channel part. This is because KL divergence does not pay attention to sorting and neglects low-probability channels. Although the large channels in KD receive more attention, the alignment value alone cannot achieve order consistency, thus losing some knowledge. The results indicate that our approach consistently enhances channel alignment, underscoring the robustness and general applicability of our method, particularly in its focused attention on low-probability channels.

**Further Ablation Study of the Hyperparameters.** To further substantiate the generalization capability and robustness of our method, we conducted comprehensive ablation studies using different combinations of coefficients. As illustrated in Fig. 8, our method consistently maintains high performance across various settings, underscoring the strong generalization ability and universal applicability of our method.

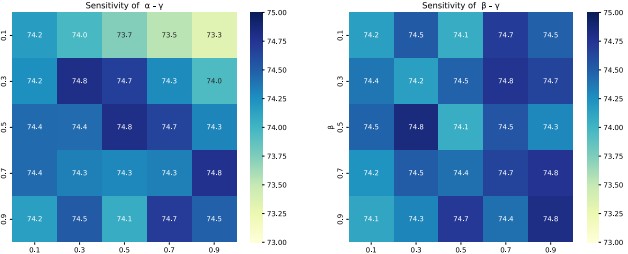

*Figure 8.* Sensitivity Analysis. **Left:** Sensitivity of $\alpha - \gamma$. **Right:** Sensitivity of $\beta - \gamma$.

## 6. Conclusion

In this paper, we investigate the optimization process of logit distillation and identify that the Kullback-Leibler divergence tends to overlook the knowledge embedded in the low-probability channels of the output. Moreover, KL divergence does not guarantee alignment between the classification results of the student and teacher models. To address this issue, we introduce a plug-and-play ranking loss based on Kendall's $\tau$ Coefficient that encourages the student model to pay more attention to the knowledge contained within the low-probability channels, while also enforcing alignment with the teacher's predictive outcomes. We provide a theoretical analysis demonstrating that the gradients of the ranking loss are less affected by channel scale and that its optimization objective is consistent with that of KL divergence, making it an effective auxiliary loss for distillation. Extensive experiments validate that our approach significantly enhances the distillation performance across various datasets and teacher-student architectures.

## Acknowledgements

This work is supported by the National Key R&D Program of China (2022YFB4701400/4701402), SSTIC Grant (KJZD20230923115106012, KJZD20230923114916032, GJHZ20240218113604008), Beijing Key Lab of Networked Multimedia.

## Impact Statement

This paper presents work whose goal is to advance the field of Machine Learning. There are many potential societal consequences of our work, none of which we feel must be specifically highlighted here.

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

# A. Appendix

### A.1. Implementation Details for Transformer

We use the AdamW optimizer and train for 300 epochs with an initial learning rate of 5e-4 and a weight decay of 0.05. The minimum learning rate is 5e-6, and the patch size is 16. We set $\alpha = 1$, $\beta = 1$, $\gamma = 0.5$, and batch size is 128. The value range of ranking loss is $[-1, 1]$ We use a single RTX4090 for CIFAR-100 and 4 RTX4090 for ImageNet.

### A.2. Ablation of Normalization

Due to the random initialization of student, the output logits will be too variant at the start of optimization and occasionally lead to gradient explosions. Therefore, we add normalization to the ranking loss to stabilize the ranking loss optimization at the very beginning. We also supplemented a set of small-scale ablation experiments that showed that the improvement in ranking performance did not come from the normalization added to the ranking loss, as shown in Tab. 10 below:

*Table 10.* Ablation of Normalization.

|         | ResNet32×4 | WRN-40-2 |
|---------|------------|----------|
| Teacher | 79.42      | 75.61    |
| Student | SHN-V1     | WRN-40-1 |
|         | 70.30      | 71.98    |
| KD              | 74.07        | 73.54        |
| KD+Ours w/o Norm | 76.38(+2.31) | 74.07(+0.53) |
| KD+Ours w Norm   | 75.98(+1.91) | 74.49(+0.95) |

### A.3. Different Forms of Ranking Loss

In the initial application of the ranking loss(Zheng et al., 2023), it is necessary to compute the gradients for two input vectors separately. However, In the scenario of distillation, the soft labels from the teacher model do not require gradients. Therefore, we propose three variants of the diff-Kendall ranking loss suitable for distillation scenarios, aimed at further exploring the role of ranking loss in distillation. Since the sample pair $(z_i^t - z_j^t)$ itself has a sign,, we can derive an equivalent form from Eq. 5:

$$\tau = \frac{\sum_i \sum_{j<i} sgn((z_i^t - z_j^t) \cdot (z_i^s - z_j^s))}{\frac{1}{2}C(C-1)} \tag{13}$$

For Eq. 5 and Eq. 13, we can similarly transform them into differential forms of ranking loss by replacing $sgn$ with $tanh$, and gradients are not computed for the output part corresponding to the teacher, which is:

$$L_\tau^{form1} = \frac{\sum_i \sum_{j<i} tanh(z_i^t - z_j^t)_{detach} \cdot tanh(z_i^s - z_j^s)}{\frac{1}{2}C(C-1)} \tag{14}$$

$$L_\tau^{form2} = \frac{\sum_i \sum_{j<i} tanh[(z_i^t - z_j^t)_{detach} \cdot (z_i^s - z_j^s)]}{\frac{1}{2}C(C-1)} \tag{15}$$

Where $()_{detach}$ means not participating in training. Further, since the gradient of the teacher's output is not required, we can directly use the sign function instead of the $tanh$ function to obtain the ranking loss:

$$L_\tau^{form3} = \frac{\sum_i \sum_{j<i} sgn(z_i^t - z_j^t)_{detach} \cdot tanh(z_i^s - z_j^s)}{\frac{1}{2}C(C-1)} \tag{16}$$

**Different Forms of Ranking Loss.** In our investigation, we examined the performance of the three forms of distillation presented in Sec. A.3, as depicted in Tab. 11. Form1 exhibited the most superior performance, followed by Form2, with Form3 trailing yet still enhancing the efficacy of the original knowledge distillation. This suggests that within the ranking loss, it is imperative to optimize considering the magnitude of the teacher's logits differences as a coefficient for the loss, rather than merely optimizing as a sign function.

*Table 11.* Different Forms of Ranking Loss. The experiments are conducted on the CIFAR-100, with 9 heterogeneous and 7 homogeneous architectures. The average Top-1 accuracy (%) is reported.

| Loss Form | KD | KD+Form1 | KD+Form2 | KD+Form3 |
|---|---|---|---|---|
| Similar Structure | 72.01 | **73.47(+1.46)** | 73.42(+1.41) | 73.38(+1.37) |
| Different Structure | 72.74 | **73.50(+0.76)** | 73.36(+0.62) | 73.05(+0.31) |

## A.4. Algorithm

---
**Algorithm 1** Plug-and-Play Ranking Loss for Logit Distillation

---
**Input:** Transfer set $\mathcal{D}$ with samples of image-label pair $\{x_n, y_n\}_{n=1}^N$, base temperature $T$, teacher $f_t$, student $f_s$, knowledge distillation Loss $\mathcal{L}_{KD}$, ranking Loss $\mathcal{L}_{RK}$, the weight of ranking Loss $\gamma$.
**Output:** Trained student model $f_s$
**for** $(x_n, y_n)$ *in* $\mathcal{D}$ **do**
  **Get** the logits of Teacher and student: $z^t = f_t(x_n)$, $z^s = f_s(x_n)$
  **Calculate** the probability with temperature: $q^t = softmax(\frac{z^t}{T})$, $q^s = softmax(\frac{z^s}{T})$

  **Get** the normalized logits of Teacher and student: $\hat{z}^t = \frac{z^t - \bar{z^t}}{std(z^t)}$, $\hat{z}^s = \frac{z^s - \bar{z^s}}{std(z^s)}$
  **Update** $f_s$ towards minimizing: $\mathcal{L}_{total} = \mathcal{L}_{KD}(q^t, q^s) + \gamma \cdot \mathcal{L}_{RK}(\hat{z}^t, \hat{z}^s)$
**end**

---

## A.5. Derivation of the KL Loss and Ranking Loss

**Derivation of the KL Divergence with Respect to Student Logits in Knowledge Distillation.**

Denote the teacher's logits as $\mathbf{z}^t = [z_1^t, z_2^t, \ldots, z_C^t]$.
Denote the student's logits as $\mathbf{z}^s = [z_1^s, z_2^s, \ldots, z_C^s]$.
Let $T$ be the temperature scaling factor used in the softmax function.

Teacher probabilities can be calculated as:

$$q_i^t = \frac{\exp\left(\frac{z_i^t}{T}\right)}{\sum\limits_{j=1}^{C} \exp\left(\frac{z_j^t}{T}\right)}, \quad \text{for } i = 1, 2, \ldots, C. \tag{17}$$

Student probabilities can be calculated as:

$$q_i^s = \frac{\exp\left(\frac{z_i^s}{T}\right)}{\sum\limits_{j=1}^{C} \exp\left(\frac{z_j^s}{T}\right)}, \quad \text{for } i = 1, 2, \ldots, C. \tag{18}$$

The loss function used in knowledge distillation is the scaled Kullback-Leibler (KL) divergence between the teacher and student probability distributions:

$$L_{KD} = T^2 \cdot \text{KL}\left(q^t \parallel q^s\right) = T^2 \sum_{i=1}^{C} q_i^t \log\left(\frac{q_i^t}{q_i^s}\right). \tag{19}$$

The derivative of the loss with respect to the student logits $z_i^s$ is:

$$\frac{\partial L_{KD}}{\partial z_i^s} = -T^2 \sum_{k=1}^{C} q_k^t \frac{\partial \log q_k^s}{\partial z_i^s}. \tag{20}$$

Taking the natural logarithm:

$$\log q_k^s = \frac{z_k^s}{T} - \log\left(\sum_{j=1}^{C} \exp\left(\frac{z_j^s}{T}\right)\right). \tag{21}$$

Compute the partial derivative:

$$\frac{\partial \log q_k^s}{\partial z_i^s} = \frac{\partial}{\partial z_i^s}\left(\frac{z_k^s}{T}\right) - \frac{\partial}{\partial z_i^s}\left(\log\left(\sum_{j=1}^{C} \exp\left(\frac{z_j^s}{T}\right)\right)\right). \tag{22}$$

Compute each term separately:

$$\frac{\partial}{\partial z_i^s}\left(\frac{z_k^s}{T}\right) = \frac{1}{T}\delta_{ik}, \quad \delta_{ik} = \begin{cases} 1, & \text{if } i = k, \\ 0, & \text{if } i \neq k. \end{cases} \tag{23}$$

$$\frac{\partial}{\partial z_i^s}\left(\log\left(\sum_{j=1}^{C} \exp\left(\frac{z_j^s}{T}\right)\right)\right) = \frac{1}{\sum_{j=1}^{C} \exp\left(\frac{z_j^s}{T}\right)} \cdot \frac{\partial}{\partial z_i^s}\left(\sum_{j=1}^{C} \exp\left(\frac{z_j^s}{T}\right)\right). \tag{24}$$

The derivative inside the sum is:

$$\frac{\partial}{\partial z_i^s}\left(\sum_{j=1}^{C} \exp\left(\frac{z_j^s}{T}\right)\right) = \exp\left(\frac{z_i^s}{T}\right) \cdot \frac{1}{T} = \frac{\exp\left(\frac{z_i^s}{T}\right)}{T}. \tag{25}$$

Therefore, the second term becomes:

$$\frac{1}{\sum_{j=1}^{C} \exp\left(\frac{z_j^s}{T}\right)} \cdot \frac{\exp\left(\frac{z_i^s}{T}\right)}{T} = \frac{1}{T}q_i^s. \tag{26}$$

Thus, the total derivative is:

$$\frac{\partial \log q_k^s}{\partial z_i^s} = \frac{1}{T}\delta_{ik} - \frac{1}{T}q_i^s = \frac{1}{T}(\delta_{ik} - q_i^s). \tag{27}$$

We now substitute $\dfrac{\partial \log q_k^s}{\partial z_i^s}$ back into the expression for the derivative of the loss:

$$\begin{aligned} \frac{\partial L_{\text{KD}}}{\partial z_i^s} &= -T^2 \sum_{k=1}^{C} q_k^t \left(\frac{1}{T}(\delta_{ik} - q_i^s)\right) \\ &= -T \sum_{k=1}^{C} q_k^t(\delta_{ik} - q_i^s). \end{aligned} \tag{28}$$

Considering:

$$\sum_{k=1}^{C} q_k^t \delta_{ik} = q_i^t, \tag{29}$$

$$\sum_{k=1}^{C} q_k^t q_i^s = q_i^s \sum_{k=1}^{C} q_k^t = q_i^s \cdot 1 = q_i^s, \quad \sum_{k=1}^{C} q_k^t = 1. \tag{30}$$

Therefore, the loss derivative simplifies to:

$$\frac{\partial L_{\text{KD}}}{\partial z_i^s} = -T \left( q_i^t - q_i^s \right). \tag{31}$$

**Derivation of the Ranking Loss with Respect to Student Logits in Knowledge Distillation.**

The Rank loss is calculated as:

$$L_{RK} = -\frac{\sum_i \sum_{j<i} tanh(k(z_i^t - z_j^t)) \cdot tanh(k(z_i^s - z_j^s))}{\frac{C(C-1)}{2}} \tag{32}$$

The derivation is calculated as:

$$\frac{\partial L_{RK}}{\partial z_i^s} = \frac{1}{\frac{C(C-1)}{2}} \sum_{j \neq i} \frac{\partial}{\partial z_i^s} \frac{1}{2} \cdot \left[ \tanh\left(k(z_i^s - z_j^s)\right) \tanh\left(k(z_i^t - z_j^t)\right) \right] \tag{33}$$

$$= \frac{1}{C(C-1)} \sum_{j \neq i} \frac{\partial}{\partial z_i^s} \left[ \tanh\left(k(z_i^s - z_j^s)\right) \tanh\left(k(z_i^t - z_j^t)\right) \right] \tag{34}$$

Denote $\phi_{ij}$ as:

$$\phi_{ij} = \tanh\left(k(z_i^s - z_j^s)\right) \tanh\left(k(z_i^t - z_j^t)\right) \tag{35}$$

Its derivative w.r.t. $z_i^s$ is:

$$\frac{\partial \phi_{ij}}{\partial z_i^s} = \left[1 - \tanh^2\left(k(z_i^s - z_j^s)\right)\right] \cdot k \cdot \tanh\left(k(z_i^t - z_j^t)\right) \tag{36}$$

Finally, the gradient of $L_{RK}$ with respect to $z_i^s$ is:

$$\frac{\partial L_{RK}}{\partial z_i^s} = -\frac{k}{C(C-1)} \sum_{j \neq i} \left[1 - \tanh^2\left(k(z_i^s - z_j^s)\right)\right] \tanh\left(k(z_i^t - z_j^t)\right) \tag{37}$$

## A.6. Derivation of the Differential Ranking Loss

In this section, we explain how we get the final form of Eq. 6. Noticed that:

$$tanh(x) = \frac{e^x - e^{-x}}{e^x + e^{-x}} = \frac{e^{2x} - 1}{e^{2x} + 1} = 1 - \frac{2}{e^{2x} + 1} \tag{38}$$

Then, we can get the final form of Eq. 6 by expanding the $tanh$ function.

$$\tau_d = \frac{\sum_i \sum_{j<i} tanh(k \cdot (z_i^t - z_j^t)) \cdot tanh(k \cdot (z_i^s - z_j^s))}{\frac{1}{2}C(C-1)} \tag{39}$$

$$= \frac{2}{C(C-1)} \cdot \sum_i \sum_{j<i} \left(1 - \frac{2}{1 + e^{2 \cdot (z_i^t - z_j^t) \cdot k}}\right) \cdot \left(1 - \frac{2}{1 + e^{2 \cdot (z_i^s - z_j^s) \cdot k}}\right) \tag{40}$$

## A.7. Does Temperature Solve the Problem of Ignoring Low-Probability Channels

Some methods (Wei et al., 2025; Cheng et al., 2025) address the issue of neglecting low-probability channels by employing temperature scaling. We examined the probability distribution of LSKD (Sun et al., 2024), an approach using adaptive temperature that has shown promising results. As shown in Fig .9, while LSKD somewhat alleviates the problem of low-probability channels, the transformed probabilities still predominantly occupy these low-probability channels. Thus, using temperature scaling does not effectively resolve the issue of neglecting low-probability channels.

*Figure 9.* probability produced by KD and LSKD

## A.8. Combining with Other Correlation Coefficients.

DIST (Huang et al., 2022) explores the use of the Pearson correlation coefficient for knowledge distillation. However, unlike our proposed method, the Pearson correlation coefficient measures the linear relationship between two variables. Our method focuses on the ranking relationship of logits, which is not strictly a linear relationship. Therefore, Kendall's $\tau$ correlation coefficient may be more suitable for capturing this non-linear ranking information. To further investigate the combined effect of using both linear and non-linear auxiliary functions, we incorporate ranking loss into DIST, which improves its performance. The results are shown in Tab. 12.

*Table 12.* Experiments on Combining DIST.

| Teacher
Student | ResNet32×4
WRN-16-2 | ResNet32×4
WRN-40-2 | WRN-40-2
SHN-V1 | VGG13
VGG8 | WRN-40-2
WRN-40-1 |
|---|---|---|---|---|---|
| DIST (Huang et al., 2022) | 75.58 | 78.02 | 76.00 | 73.80 | 74.73 |
| DIST+Ours | 75.85(+0.27) | 78.54(+0.52) | 76.23(+0.23) | 74.06(+0.26) | 74.86(+0.13) |

## A.9. More ImageNet Results

We further augment CTKD (Li et al., 2023) and NKD (Yang et al., 2023a) with the proposed ranking loss and evaluate them on the ImageNet dataset; the results are reported in Tab. 13.

*Table 13.* The top-1 and top-5 accuracy (%) on the ImageNet validation set. The teacher and student are ResNet34 and ResNet18

| Method | Top-1 Acc. | Top-5 Acc. |
|---|---|---|
| AT (Zagoruyko & Komodakis, 2017) | 70.69 | 90.01 |
| OFD (Heo et al., 2019) | 70.81 | 89.98 |
| CRD (Tian et al., 2019) | 71.17 | 90.13 |
| CTKD (Li et al., 2023) | 71.32 | 90.27 |
| CTKD+Ours | 71.68(+0.36) | 90.30(+0.03) |
| NKD (Yang et al., 2023a) | 71.96 | 90.27 |
| NKD+Ours | 72.06(+0.10) | 90.49(+0.22) |

