# OpenReview forum: "Enhancing Logits Distillation with Plug&Play Kendall's  $\tau$ Ranking Loss"
_ICML.cc/2025/Conference — ICML 2025 poster_

### Official Review · Reviewer_wbp7 · 2025-02-18

**Overall Recommendation:** 3

**Summary:**

This paper presents a method to enhance logits distillation by incorporating a ranking loss based on Kendall’s τ coefficient. The main finding is that the conventional knowledge distillation approach, which relies heavily on KL divergence, often neglects smaller logit channels that contain valuable information. The authors introduce a ranking loss that ensures the order consistency between the teacher and student logits. The main algorithmic idea is to combine the traditional KL divergence loss with the proposed ranking loss, allowing the student model to not only align with the teacher's probability distribution but also maintain the ordinal relationship among logits.
## update after rebuttal
I have carefully read the rebuttal and the other comments. The rebuttal provide the new experimental results and the explanation of the motivation and clearly illustrate the scenario that low-probability channels receiving smaller gradients under the KL divergence. The rebuttal also provide the role of the ranking loss for the the gradients of low-probability labels. Some of my concerns are well solved. Therefore, I change my score. However, some issues like the motivation of using ranking loss, the innovation of the ranking loss, are not clealy discussed.

**Claims And Evidence:**

1.	The paper merely states that using KL divergence for distillation loss ignores the gradient information of low-probability labels, but this viewpoint lacks any theoretical support. In Equation 3, $q_i^t$ merely indicates that the gradient information for small labels will be small, which does not imply that they are ineffective during learning. Furthermore, low-probability labels themselves should not dominate the information in the distillation process.
2.	The proof of consistency between the ranking loss and the KL divergence optimization objective has only been preliminarily explored from the perspective of the optimal solution space, without a thorough analysis of their relationship in the context of complex models and data. This insufficient analysis cannot fully explain why the ranking loss can effectively assist in knowledge distillation. The presentation merely elaborates on the advantages of the ranking loss, lacking more rigorous mathematical proof and theoretical support, and fails to provide an in-depth explanation of the internal mechanisms of the proposed method.

**Essential References Not Discussed:**

These two works should be selected as comparison methods in the experimental setup.
[1] A unified approach with normalized loss and customized soft labels
[2] DOT: A Distillation -Oriented Trainer

**Experimental Designs Or Analyses:**

There are three issues with the experimental design section of the paper:
1. The experiments conducted in the paper are insufficient, as many teacher-student combinations have not been covered, such as Res56/Res20, wrn40-2/wrn16-2, vgg13/mbv2, res32x4/shv1, etc.
2. In the experiments on ImageNet, the top-5 accuracy results are missing, and the experimental results for DKD+, CTKD+, and MLKD+ are also lacking.
3. The advantages compared to the logit-standard method are too insignificant, with differences of only 0.1-0.2.

**Methods And Evaluation Criteria:**

The evaluation criteria employed in the paper include accuracy metrics (top-1 and top-5) on benchmark datasets like CIFAR-100, ImageNet, and MS-COCO. These datasets and evaluation are widely used in KD.

**Other Comments Or Suggestions:**

1. Conduct an in-depth analysis of the gradient characteristics of the ranking loss based on Kendall’s τ coefficient, for instance, by rigorously mathematically deriving and proving its stability with minimal influence from channel scale across different model structures and data distributions.
2. Utilize visualization  (such as feature importance visualization, label information visualization, etc.) to conduct an in-depth analysis of the specific role that the inter-class relationship information provided by the ranking loss plays in knowledge distillation.

**Other Strengths And Weaknesses:**

My major concern is the rationality of the motivation.
1. Although the KL divergence has the issue of neglecting low-probability channels, the paper does not clearly elaborate on the rationality of introducing a ranking loss in knowledge distillation. Figure 1 seems more like a hypothetical scenario and lacks real-case analysis.
2. Second, the differences in experimental results compared to other SOTA methods are too small to sufficiently demonstrate the effectiveness of the proposed method.
3. Third, there is no detailed analysis of the impact on model training time and computational resource consumption after adding the ranking loss.

**Questions For Authors:**

1. The paper mentions that the gradient of the ranking loss is less affected by channel scale and that its optimization objective is consistent with the KL divergence. Can we theoretically analyze how the ranking loss enhances the classification ability of the student network?
2. There are various types of ranking losses. What are the reasons for choosing Kendall’s τ? Can we add comparisons with other ranking losses in the experimental section?
3. In the appendix, could you provide some derivation details on how Eq. 28 is derived to Eq. 31?

**Relation To Broader Scientific Literature:**

1. The paper proposes an auxiliary ranking loss based on Kendall's τ coefficient to mitigate the neglect of low-probability channels by the Kullback-Leibler (KL) divergence loss. This ranking loss is designed to focus on the alignment of logits between the teacher and student models, taking into account both high- and low-probability channels. This contribution builds upon and extends the existing knowledge distillation framework by incorporating a new loss function that addresses a specific limitation of the KL divergence.
2. The paper demonstrates that the proposed ranking loss can be used as a plug-and-play auxiliary function in various distillation tasks, which is similar to the setting in Logit-standard KD.

**Theoretical Claims:**

The paper does not elaborate on any theory. In Section 4.2, it discusses the advantages of the ranking loss from three different perspectives. The proofs for the first two perspectives do not contain typos, but the analysis for the third perspective is clearly incorrect. Knowledge distillation aims to enhance the generalization of the model through the KL divergence and ensures the correctness of classification through cross-entropy loss. Assigning a larger weight to the distillation loss is precisely to ensure that the logits of the student network are closer to those of the teacher network. This analysis does not demonstrate that the ranking loss can help the student network achieve better classification. This part of the analysis should instead focus on how the introduction of the ranking loss can help the student network better ensure that the optimal label dominates, reducing the risk of low-probability labels exceeding the optimal label during the learning process.

---

> ### Author Rebuttal · Authors · 2025-04-01
>
> We sincerely thank you for the in-depth feedback! We highly value each of your comments, and all your concerns are addressed point by point:
>
> ---
>
> **Q1: Claims And Evidence**
> 1. **Small gradient**: We would like to clarify that while small gradient information does not necessarily imply ineffectiveness in the learning process, it is indeed prone to being overlooked. Our goal is for the model to learn more from the distribution of low-probability channels rather than ignoring them. Furthermore, we do not treat low-probability channels as dominant information; instead, we ensure that the model learns more consistently ranked logits under the constraint of KL divergence, thereby better capturing the knowledge hidden within low-probability channels.
> 2. **Optimal solution space**: We would like to clarify that the optimal solution space primarily to illustrate the compatibility between ranking loss and KL, meaning that optimizing ranking loss does not significantly interfere KL. However, this is not the main reason why ranking loss improved distillation performance.
>
> ---
>
> **Q2: Theoretical Claims**
> We would like to clarify that correct ranking inherently encompasses the notion of correct classification. Notably, the teacher model generally achieves accurate classification on the training set; therefore, ensuring consistency in the ranking order between the student and the teacher, to some extent, also guarantees correct classification for the student. Consequently, the ranking loss serves to enhance the classification accuracy of the student model.
>
> ---
>
> **Q3： The rationality of the motivation**
>
> 1. **The rationality of introducing the ranking loss**: The motivation for introducing ranking loss is to help the student model better capture the inter-class information of small channels and help the student model classify accurately without affecting the optimization of KL divergence. These are the three main perspectives we discuss in the paper. Figure 1 is not a hypothetical scenario, it has also been mentioned in works such as LSKD [1].
>
> 2. **Limited improvement**: Our average improvement is 0.85, which we believe is effective. Considering the capacity of the student model and its gap with the teacher model, we suggest that our method effectively improves the performance of distillation.
>
> 3. **Time comsumption**: **Table T4.1** shows the distillation optimization time with and without ranking loss. It can be seen that ranking loss only brings a small extra time consumption.
>
> **Table T4.1**: Additional Computation Evaluation of Ranking Loss. The full training time is reported as the evaluation metric
>   | Dataset | CIFAR-100|
>   |:---:|:---:|
>   | KD      | 0.93 Hours |
>   | KD+Ours | 1.06 Hours |
>
> ---
>
> **Q4: How the ranking loss enhances the classification ability of the student network**
> Intuitively, a tiger should be more like a cat than a fish. This relationship between channels can help the model better capture generalization features. According to previous work [2], the essence of distillation is to help the student model to learn features from more views to improve generalization. Therefore, ranking loss can help the student model generalize more effectively. Ranking loss enhances the training classification accuracy of the student model and better understanding the knowledge of small channels without affecting the optimization of KL divergence, thereby improving the generalization ability of the student model.
>
> ---
>
> **Q5: Other types of ranking losses.**
> Goodman Kruskal's Gamma can be understood as Kendall's with ties ignored, but these ties have no gradients when optimizing Kendall's taus, so they are essentially the same when used for optimization. Considering the form of Spearman’s rank correlation coefficient, it is difficult to convert it into a differentiable form. Considering the above reasons, using the Kendall coefficient tau as a ranking loss is a suitable choice.
>
> ---
>
> **Q6: Derivation details**
> Based on equation 23, we can simply get equation 29. Considering that the sum of the probability output by the model is 1, we can get equation 30. Expanding equation 28 and substituting it into equations 29 and 30, we can get equation 31. We will further clarify the steps here in the revision.
>
> [1] Logit Standardization in Knowledge Distillation
>
> [2] Towards Understanding Ensemble, Knowledge Distillation and Self-Distillation in Deep Learning

---

> > ### Comment · Reviewer_wbp7 · 2025-04-02
> >
> > Thanks for your effort to provide the rebuttal. However, I still have a concern about the motivation since the KL divergence does not actually ignore the gradients of low-probability labels; rather, it distills knowledge based on the overall similarity between the two distributions. It seems unreasonable to separate and analyze the KL divergence in this way. Second, the response does not explain why ranking can address the gradients of low-probability labels or why it is less affected by the channel scale. Third, in the third and fourth questions, the improvement in classification still mainly relies on the cross-entropy loss function, while the purpose of the distillation loss is to generalize knowledge. I still have differing opinions regarding the motivation set by the authors.

---

> > > ### Author Response · Authors · 2025-04-07
> > >
> > > We sincerely thank you for the in-depth feedback!
> > >
> > > ---
> > >
> > > **Q1. About Motivation:**
> > >
> > > - As shown in **Figure 3 of the main paper and Section A.6 of the Appendix**, we provide both visualization and theoretical derivation demonstrating that low-probability channels receive smaller gradients under the KL divergence. Although KL divergence can capture very general knowledge through distribution learning, making it the most important loss in distillation tasks, the proposed ranking loss demonstrates stronger capabilities in capturing low-probability channels and inter-class relationships. This serves as a complement to KL divergence. **Figures 3 and 4 in the paper** illustrate that the ranking loss performs better in these aspects and effectively helps further reduce the distillation loss and improve performance.
> > >
> > > ---
> > >
> > > **Q2. Why ranking is less affected by the channel scale:**
> > >
> > > - In **Section A.6 of the Appendix**, we provide the derivation of the gradients for the ranking loss, which is given by $ \frac{\partial  L_{RK}}{\partial z^s_i} = -\frac{k}{C(C - 1)} \sum_{j \ne i} \left[ 1 - \tanh^2\left( k (z^s_i - z^s_j) \right) \right] \tanh\left( k (z^t_i - z^t_j) \right)  $ . The tanh function and an appropriate steepness parameter control the magnitude of the channel difference term, making it less sensitive to channel scale. In contrast, the gradient provided by KL divergence, given by $\frac{\partial L_{\text{KD}}}{\partial z^{s}_i} = -T \left( q^{t}_i - q^{s}_i \right)$ , includes a difference term that makes it more sensitive to channel scale. This is also illustrated in **Figure 3 of the paper**, which visualizes the gradients obtained for channels of different sizes.
> > >
> > > ---
> > >
> > > **Q3. Questions about classification:**
> > >
> > > - In fact, both KL divergence and ranking loss significantly contribute to the improvement of classification. The cross-entropy loss only constrains the target class, while the soft labels provided by the teacher contain rich inter-class information, which not only aids generalization but also facilitates global optimization for classification. KL divergence and ranking loss enhance classification performance by capturing such information.
> > >
> > > - Our further experiments also demonstrate that students trained with ranking loss achieve higher accuracy and better generalization ability, supporting our hypothesis that ranking can capture information relevant to classification.
> > >
> > > - Additionally, while cross-entropy loss aims for a predicted probability of 1 for the target class, KL divergence aims for the predicted probabilities of all classes to align with the soft labels provided by the teacher. There is an inherent mismatch in their solution spaces, which often necessitates assigning a smaller weight to cross-entropy during distillation. In contrast, ranking loss operates in a space aligned with KL divergence and explicitly preserves ranking consistency, thereby maintaining distillation efficiency while enhancing the consistency of decision boundaries, ultimately improving classification.

---

### Official Review · Reviewer_Zvoo · 2025-02-26

**Overall Recommendation:** 3

**Summary:**

This paper introduces a plug-and-play ranking loss based on Kendall’s τ to mitigate two drawbacks of KL-based logit distillation: the neglect of low-probability channels and getting stuck in suboptimal points. By aligning channel-wise rankings between teacher and student, it leverages inter-class relationships and balances attention across all channels. The loss is theoretically shown to be compatible with KL divergence—sharing the same optimal solution—yet more robust to scale. Experiments on CIFAR-100, ImageNet, and MS-COCO, using both CNNs and ViTs, confirm that adding this ranking loss consistently enhances performance over baselines.

**Claims And Evidence:**

Yes

**Essential References Not Discussed:**

To the best of my knowledge, I think the authors have considered the relevant works related to their study.

**Experimental Designs Or Analyses:**

Yes. The authors compared their method with current state-of-the-art approaches on common visual distillation tasks using CIFAR-100, COCO, and ImageNet. I find the experiments to be quite comprehensive overall. A potential shortcoming maybe is that they did not evaluate the method’s effectiveness in the increasingly popular large-model distillation settings.

**Methods And Evaluation Criteria:**

Yes

**Other Comments Or Suggestions:**

See above

**Other Strengths And Weaknesses:**

Strengths:

1. This paper is well-written and easy to follow.
2. The experiments are extensive and conducted on CIFAR-100, ImageNet, and COCO datasets. Additionally, more visual analyses are provided to explain the effectiveness of the proposed method.

Weaknesses：
1. The font in Figure 1 appears to be very small.
2. The proposed motivation does not seem very reasonable. I have a different opinion from the authors regarding why KL divergence would overlook the matching of low-probability channels. Intuitively, KL divergence seems to assign different learning weights to different class channels, with the target class having a higher weight, making it more important. This appears to be a reasonable arrangement. Although the authors provide an example in Figure 1, I do not believe this phenomenon occurs frequently.
3. Following the previous question, if time permits, could the authors compare the performance of using only the proposed rank loss versus using only KL divergence? This would provide a more intuitive evaluation of the effectiveness of the proposed method.

**Questions For Authors:**

I don’t quite understand lines 183-185, which state: “The gradient of a logit channel primarily depends on the difference between its rank and the target rank, effectively harnessing the knowledge from smaller channels.” From Eq.9 and Eq.10, I don’t seem to arrive at this conclusion. Could the author provide further clarification?

**Relation To Broader Scientific Literature:**

This study analyzes the deficiencies in a series of previous logit-based methods based on KL divergence and proposes a novel auxiliary ranking loss based on Kendall’s τ Coefficient to mitigate the aforementioned issues.

**Theoretical Claims:**

Yes

---

> ### Author Rebuttal · Authors · 2025-04-01
>
> We sincerely thank you for the in-depth feedback! We highly value each of your comments, and all your concerns are addressed point by point:
>
> ---
>
> **Q1. Font Size in Figure 1:**
>
> - Thank you very much for your feedback regarding the formatting of our paper. We will will use appropriate font sizes in the future version of the manuscript. We appreciate your attention to detail.
>
> ---
>
> **Q2. The Motivation Seems Not Reasonable: Intuitively, KL divergence seems to assign different learning weights to different class channels, with the target class having a higher weight, making it more important.**
>
> - In the distillation task, the cross-entropy loss computed with the one-hot hard labels of the target class has already emphasized its significance. The soft labels output by the teacher model differ from the one-hot hard labels, as they contain richer inter-class relationship knowledge. While the Kullback-Leibler (KL) divergence captures this information by constraining the distribution, it tends to overlook the low-probability channels. In contrast, the proposed ranking loss exhibits a stronger capability in capturing relational information among low-probability channels, thereby providing complementary support to the KL divergence.
>
> ---
>
> **Q3. The Motivation Seems Not Reasonable: The phenomenon in Figure 1 does not occur frequently.**
>
> - Figure 1 presents a toy case designed to illustrate the potential suboptimal optimization behavior of the Kullback-Leibler (KL) divergence. A similar suboptimal scenario is also demonstrated in **Figure 2 of the LSKD [1] paper**. To further validate whether the proposed ranking loss can address the potential suboptimality in the KL divergence, we visualize the loss landscape in **Figure 6 of our manuscript**. We observe that when only the KL divergence is used, suboptimal points exist near the global optimum, while the introduction of the ranking loss effectively mitigates this issue.
>
> [1] Logit Standardization in Knowledge Distillation. CVPR 2024 Highlight
>
> ---
>
> **Q4. Performance Comparison of using only Ranking Loss versus using only KL Divergence:**
>
> - We are currently conducting comparative experiments using only the ranking loss and only the KL divergence. We will promptly update our response once the results are obtained.
> - In fact, the proposed ranking loss serves as an auxiliary function to the KL divergence, designed to guide the KL divergence in avoiding suboptimal solutions and capturing inter-class relationship information. The ranking loss focuses solely on aligning the order of channels and imposes minimal constraints on channel values and distributions. Therefore, using only the ranking loss typically does not yield satisfactory distillation performance.

---

> > ### Comment · Reviewer_Zvoo · 2025-04-03
> >
> > Thank you for your detailed rebuttal and the rigorous experimental efforts. I will decide whether to revise my score after the authors address the additional experiments mentioned in Q4.
> > Additionally, the core idea of this work appears highly similar to LSKD. However, as shown in Table 1, the proposed method’s performance is notably inferior to that of LSKD. Alternatively, given the tight timeline, I wonder if integrating the authors’ approach with LSKD could yield further performance improvements.

---

> > > ### Author Response · Authors · 2025-04-06
> > >
> > > We sincerely thank you for the in-depth feedback!
> > >
> > > ---
> > >
> > > **Q1. Additional Experiments:**
> > >
> > > - We separately test distillation using only the KL divergence and only the proposed ranking loss, with the results presented in **Table T3.1**. We observe that using only the ranking loss can achieve comparable results than using the KL divergence.
> > >
> > > *Table T3.1: Ablation on KL.*
> > >   | Teacher -> Student | WRN-40-2 -> WRN-40-1 | VGG13 -> VGG8| ResNet32×4 -> SHN-V1 |  ResNet32×4 -> WRN-40-2 | WRN-40-2 -> SHN-V1 |
> > >   | :---: | :---: | :---: | :---: | :---: | :---: |
> > >   | Only KL           | 73.54 | 72.98 | 74.07 | 77.70 | 74.83 |
> > >   | Only Ranking Loss | 74.00 | 73.72 | 75.60 | 78.04 | 75.81 |
> > >   | KL + Ranking Loss | 74.49 | 74.14 | 75.98 | 78.50 | 76.13 |
> > >
> > > ---
> > >
> > > **Q2. Combination with LSKD:**
> > >
> > > - The LSKD results reported in **Table 1 of the paper** are for MLKD+LSKD, while MLKD+Ours outperformed LSKD in 7 out of 9 experiments, with an average performance improvement of 0.36.
> > > -  Additionally, LSKD is a method that optimizes the temperature of the KL divergence to enhance its generalization ability in aligning distributions, whereas the ranking loss focuses more on aligning channel order to provide additional inter-class information. Their core ideas are fundamentally different. Moreover, unlike LSKD, which modifies the KL divergence, the ranking loss is a plug-and-play auxiliary loss.
> > > - We incorporate LSKD into our proposed method and observed a performance improvement, and the ablation results are presented in **Table T3.2**.
> > >
> > > *Table T3.2: Ablation on LSKD.*
> > >   | Teacher -> Student | WRN-40-2 -> WRN-40-1 | ResNet32×4 -> SHN-V1 |  ResNet32×4 -> WRN-40-2 | WRN-40-2 -> SHN-V1 |
> > >   | :---: | :---: | :---: | :---: | :---:
> > >   | KD+LSKD      | 74.37 | 75.12 | 77.92 | 75.53 |
> > >   | KD+Ours      | 74.49 | 75.98 | 78.50 | 76.13 |
> > >   | KD+LSKD+Ours | 74.95 | 76.12 | 78.74 | 76.15 |

---

### Official Review · Reviewer_vqaG · 2025-03-11

**Overall Recommendation:** 3

**Summary:**

This paper points out two drawbacks of KL divergence in knowledge distillation: (1) it is often prone to suboptimal points; (2) it overlooks low-probability channels. The authors use Kendall’s $\tau$ coefficient to mitigate these issues and better model inter-class relationships. Experiments on CIFAR-100, ImageNet, and COCO across various architectures demonstrate improved performance compared to baselines.

## Update after rebuttal

I have carefully read the authors' rebuttal as well as reviews provided by other reviewers.

- Some of my concerns have been addressed.

The authors reasonably explain that KL tends to ignore low-probability channels due to their small gradients, but this point should be further elaborated in the main paper. Moreover, the authors have provided additional experimental results.

- **Nonetheless, there remains a critical concern that has not been addressed.**

The motivation for introducing the ranking loss—enabling the student model to better capture inter-class relationships (IRs)—is highly related to existing methods (e.g., [WKD]). However, **the authors failed to discuss connections with and differences from WKD.**

Unlike using logit ranking as qualitative information in this paper, WKD quantitatively models IRs based on feature-level statistics and employs the Wasserstein Distance (WD) to measure the distance between discrete probability distributions across all classes. *From a theoretical standpoint, WKD may offer a more principled approach than the proposed ranking loss.*

Therefore, the authors are required to explicitly discuss methodological distinctions and offer theoretical insights in the main paper. Furthermore, it is important to empirically evaluate the individual performance of ranking loss and WKD and their potential complementarity. Such discussions would help further substantiate the soundness and effectiveness of the proposed approach.

[WKD] Wasserstein Distance Rivals Kullback-Leibler Divergence for Knowledge Distillation. NeurIPS, 2024.

- **Due to the unresolved concern, my actual evaluation lies between "weak reject" and "weak accept". However, as no such score under the current review criteria, I select "weak accept".**

**Claims And Evidence:**

1.	According to Eq.1, the authors claim that KL overlooks low-probability channels as the teacher's probability serves as a weighting factor. However, this claim is questionable. The scale of KL is influenced by both the teacher’s probability and the logarithmic term. Could the authors clarify the specific role of the logarithmic term in KL?

2.	In Lines 77-80 (R1), the authors claim that low-probability channels will receive smaller gradients during optimization. However, according to Eq.5, the gradient appears to be influenced by the difference of bewteen teacher and student,  and independent of the channel values. The analysis of Eq.5 seems to contradict the claim in R1.

**Essential References Not Discussed:**

The core contribution of this paper lies in utilizing rank loss to measure the inter-class relationships. In contrast, WKD employs WD to capture inter-class relationships (refer to “Experimental Designs or Analyses”). Rank loss focuses solely on class ranking, while WKD provides a quantitative measure of inter-class relationships. The authors should further discuss distinctions and connections with related works.

**Experimental Designs Or Analyses:**

1. The current experimental design on ImageNet and COCO could be further refined. The current baseline, KD, has a relatively weak performance in the context of knowledge distillation. To provide a more comprehensive evaluation of the proposed method, the authors should conduct experiments on stronger KL-based baselines and analyze performance changes, such as [NKD], [CTKD], and [WTTM].

[NKD] From Knowledge Distillation to Self-Knowledge Distillation. ICCV, 2023.

[CTKD] Curriculum Temperature for Knowledge Distillation. AAAI, 2023.

[WTTM] Knowledge Distillation based on Transformed Teacher Matching. ICLR, 2024.

2. The paper lacks a comparison with relevant SOTA methods. For instance, [WKD] also leverages inter-class relationships for distillation and employs Wasserstein Distance (WD) instead of KL as a measurement. A comparison with such methods would help better highlight the advantages of the proposed approach.

[WKD] Wasserstein Distance Rivals Kullback-Leibler Divergence for Knowledge Distillation. NeurIPS, 2024.

**Methods And Evaluation Criteria:**

Yes.

**Other Comments Or Suggestions:**

There are some typos, including the incorrect citation of Fig. 5 in Line 378 (right column). The authors should carefully review and correct these issues.

**Other Strengths And Weaknesses:**

1. The idea of introducing rank loss in knowledge distillation to overcome the limitations of KL and provide inter-class relationships is interesting.

2. The visualizations are well-executed, which is helpful to enhance the overall understanding of the paper.

**Questions For Authors:**

1.	Why does KL overlook low-probability channels? Why low-probability channels will have smaller gradients? The current theoretical analysis presented by the authors is unconvincing. A more rigorous theoretical or experimental analysis should be provided.

2.	The discussion on ranking loss is rather limited. Besides Kendall’s τ, other ordinal data measures, such as Spearman’s rank correlation coefficient and Goodman & Kruskal’s Gamma, could also be considered. It is important to analyze and compare the performance of these metrics.

**Relation To Broader Scientific Literature:**

It seems there is no broader scientific literature for this paper.

**Theoretical Claims:**

N/A

---

> ### Author Rebuttal · Authors · 2025-04-01
>
> We sincerely thank you for the in-depth feedback! We highly value each of your comments, and all your concerns are addressed point by point:
>
> ---
>
> **Q1. The scale of KL is influenced by both the teacher’s probability and the logarithmic term. Could the authors clarify the specific role of the logarithmic term in KL?**
>
> - The logarithmic term is derived from maximum likelihood, but the coefficient of the teacher is independent of the log term. As shown in the formula
> $$K L(P | Q)=\sum p(x) \log \frac{p(x)}{q(x)}=\underbrace{-\sum p(x) \log (q(x))}_1+\underbrace{\sum p(x) \log (p(x))}_2$$
> the first part is a constant, while the second part is the optimizable component. The teacher's probability, acting as the coefficient of the log term, leads to the optimization process ignoring channels where the teacher's output probability is small. The neglect of low-probability channels by the KL divergence can also be explained at the gradient level. We have provided a detailed explanation in Figure 3 of the paper and Section A.6 of the appendix to validate our observation that small channels often receive smaller gradients.
>
> ---
>
> **Q2. The analysis of Eq.5 seems to contradict the claim in R1:**
>
> - In fact, lines 77-80 (R1) explain that when using KL divergence, low-probability channels receive smaller gradients during the optimization process. Equation 5 represents our proposed ranking loss, which we designed to be independent of channel values to ensure that low-probability channels are not overlooked. Therefore, the fact that the gradient of Equation 5 is independent of channel values does not conflict with R1; rather, this is precisely the intended goal of our design.
>
> ---
>
> **Q3. The current experimental design on ImageNet and COCO could be further refined:**
>
> - We have individually integrated the proposed ranking loss into NKD [1], CTKD [2], and WTTM [3], and conducted comparative experiments on ImageNet. Notably, CTKD+Ours achieved results surpassing the baseline despite being trained for 23 fewer epochs than CTKD. The experiments for WTTM are still ongoing, and we will promptly update our response once the results are obtained. The results are presented in **Table T2.1**.
>
> *Table T2.1: Experiments on ImageNet.*
>   | ResNet34 -> ResNet18 | Acc@1 | Acc@5 |
>   | :---: | :---: | :---: |
>   | NKD[1]      | 71.96 | - |
>   | NKD[1]+Ours | 72.06 | 90.57 |
>   | CTKD[2] (120epochs)      | 71.32 | 90.27 |
>   | CTKD[2]+Ours(97epochs) | 71.43 | 90.29 |
>
> - For object detection on COCO, WTTM does not provide relevant experiments, and the NKD paper only includes experiments related to the self-supervised method USKD, which differs from our experimental setup. Therefore, we compared our method with CTKD, and the results are presented in T**able T2.2**.
>
> *Table T2.2: Experiments on COCO.*
>   | ResNet50 -> MN-V2 | AP | AP50 | AP75 |
>   | :---: | :---: | :---: | :---: |
>   | CTKD[2] | 31.39 | 52.34 | 31.35 |
>   | Ours    | 31.99 | 53.80 | 33.37 |
>
> ---
>
> **Q4. The paper lacks a comparison with relevant SOTA methods:**
>
> - In fact, we have already compared our method with LSKD [5], a highlight method from CVPR 2024, in the paper. We further add a performance comparison with WKD[4] in Table T2.3.
>
> *Table T2.3: Comparison with WKD.*
>   | Teacher -> Student | WRN-40-2 -> WRN-40-1 | ResNet32×4 ->ResNet8×4 | VGG13 -> VGG8 | WRN-40-2 -> SHN-V1 | ResNet50 -> MN-V2 |
>   | :---: | :---: | :---: | :---: | :---: | :---: |
>   | WKD-L[4] | 74.84 | 76.53 | 75.09 | 76.72 | 71.10 |
>   | Ours     | 76.08(+1.24) | 77.25(+0.72) | 75.35(+0.26) | 77.87(+1.15) | 71.66(+0.56) |
>
> ---
>
> **Q5. Typos:**
>
> - We are sorry that we could not locate any citations in Figure 5 or line 378, nor did we find any incorrect citations in Table 5. If there are any errors, we would greatly appreciate it if you could point them out to us.
>
> ---
>
> **Q6. Why does KL overlook low-probability channels? Why low-probability channels will have smaller gradients?**
>
> - In Section A.6 of the appendix, we provide a detailed derivation of the gradients provided by the KL divergence. The gradient obtained by a specific channel is related to the difference between the student and teacher values in that channel. Since the values of the student and teacher in low-probability channels are on a smaller scale, the gradients obtained are also smaller. This is the underlying reason why KL divergence tends to overlook small channels. In Figure 3 of the paper, we illustrate the gradients provided by KL divergence for channels of different sizes, which also demonstrates that low-probability channels generally receive smaller gradients.
>
> ---
>
> **Q7. Discussions on other ordinal data measures:**
>
> - We are currently experimenting with replacing the Kendall correlation coefficient with Spearman’s rank correlation coefficient and Goodman & Kruskal’s Gamma for ablation studies. The experiments are still ongoing, and we will promptly update our response with the results as soon as they are available.

---

> > ### Comment · Reviewer_vqaG · 2025-04-03
> >
> > Thank you for the detailed rebuttal, which I have carefully reviewed. Some of my concerns have been addressed. However, a few issues remain:
> >
> > First, the response in Q1 remains unclear. Contrary to the authors' explanation, I believe the first part of the equation is the optimizable component, while the second part is a constant. However, this does not justify why the optimization process ignores channels where the teacher's output probability is small. Since the scale of the first term is influenced by both the teacher and student distributions, the current explanation appears unconvincing.
> >
> > Second, on the ImageNet dataset, the proposed ranking loss yields only a marginal improvement of 0.1 when applied to strong KL-based distillation methods (e.g., NKD and CTKD). Such a limited gain raises concerns about the method’s practical impact and significance.

---

> > > ### Author Response · Authors · 2025-04-07
> > >
> > > We sincerely thank you for the in-depth feedback.
> > >
> > > ---
> > >
> > > **Q1. Why the optimization process ignores channels where the teacher's output probability is small:**
> > >
> > > - We sincerely apologize for mistakenly reversing the first and second items of Q1 in our previous response. In fact, as mentioned in line 51 of our paper, the analysis of the weight term in the KL divergence formula is intuitive, and we understand your confusion. Therefore, **we would like to emphasize once again that we have provided the detailed derivation of the gradients provided by the KL divergence in Appendix A.6 of the paper. Additionally, Figure 3 in the paper visualizes the gradients obtained by channels at different scales, theoretically and experimentally explaining why the KL divergence tends to overlook low-probability channels.** Specifically, the gradient provided by the KL divergence is $\frac{\partial L_{\text{KD}}}{\partial z^{s}_i} = -T \left( q^{t}_i - q^{s}_i \right)$, which is influenced by the difference in channel values between the teacher and student outputs, making it sensitive to the scale of channel values. For low-probability channels (especially in the later stages of optimization), since both the teacher and student outputs have small channel values, the gradient becomes smaller due to the reduced difference in channel values, causing them to be neglected by the KL divergence.
> > >
> > > ---
> > >
> > > **Q2. The ImageNet Results:**
> > >
> > > - The previously reported results for CTKD+Ours were based on incomplete training. We have now provided the fully trained results for CTKD+Ours, as well as the previously missing results for WTTM+Ours, which are presented in **Table T2.4**. The average performance improvement across the three baselines is 0.26.
> > >
> > > *Table T2.4: Experiments on ImageNet.*
> > >   |  |  ResNet34 -> ResNet18 |
> > >   | :---: | :---: |
> > >   | CTKD (120epochs)      | 71.32 |
> > >   | CTKD+Ours(120epochs) | 71.68(+0.36) |
> > >   | WTTM      | 72.19 |
> > >   | WTTM+Ours | 72.51(+0.32) |
> > >
> > > ---
> > >
> > > **Q3. Discussions on other ordinal data measures:**
> > >
> > > - The missing experiment for Q7 in our previous response has been supplemented in **Table T2.5**. It is worth noting that we found Goodman & Kruskal’s Gamma to be equivalent to our Kendall correlation coefficient. The Kendall correlation coefficient is calculated as (C - D) / N_pairs, while Goodman & Kruskal’s Gamma is calculated as (C - D) / (C + D), where C represents the number of concordant pairs and D represents the number of discordant pairs. In our Kendall coefficient, N_pairs is the sum of concordant and discordant pairs, which is equivalent to (C + D).
> > >
> > > *Table T2.5: Comparison with other coefficient.*
> > >   | Teacher -> Student | WRN-40-2 -> WRN-40-1 | VGG13 -> VGG8| ResNet32×4 -> SHN-V1 |  ResNet32×4 -> WRN-40-2 | WRN-40-2 -> SHN-V1 |
> > >   | :---: | :---: | :---: | :---: | :---: | :---: |
> > >   | Only KL           | 73.54 | 72.98 | 74.07 | 77.70 | 74.83 |
> > >   | +Spearman's Rank  | 73.79 | 73.79 | 75.51 | 78.05 | 75.14 |
> > >   | +Our Ranking Loss | 74.49 | 74.14 | 75.98 | 78.50 | 76.13 |

---

### Official Review · Reviewer_Za63 · 2025-03-13

**Overall Recommendation:** 4

**Summary:**

This paper proposes an auxiliary ranking loss based on Kendall’s tau coefficient to improve knowledge distillation by addressing the limitations of KL divergence. Traditional KL-based distillation struggles with optimization challenges and tends to overlook low-probability channels, leading to suboptimal performance. The proposed ranking loss provides inter-class relationship information and balances attention across all channels while maintaining consistency with KL divergence. It is a plug-and-play module compatible with various distillation methods. Extensive experiments on CIFAR-100, ImageNet, and COCO datasets demonstrate its effectiveness across different CNN and ViT teacher-student architectures.

**Claims And Evidence:**

Yes, the claims have been supported by clear and convincing evidences.

**Essential References Not Discussed:**

A contribution of the proposed method is to improve the object detection performance via distillation, and it is suggested to discuss several more techniques of detection improvement such as [1-2].

[1] Dynamic head: Unifying object detection heads with attentions, CVPR 2021.
[2] Rethinking image restoration for object detection, NeurIPS 2022.

**Experimental Designs Or Analyses:**

Experimental designs and analyses are valid.

**Methods And Evaluation Criteria:**

Yes, the proposed methods and evaluation criteria make sense.

**Other Comments Or Suggestions:**

It is recommended to revise the functions and equations to ensure proper formatting, condensing them into a single line where possible or adjusting line breaks appropriately, particularly for Equations (6) and (10).

**Other Strengths And Weaknesses:**

Strengths:

- The idea of emphasizing the ranking information of logits through a ranking loss is both innovative and impactful. By explicitly modeling the relative order of predictions, the proposed approach enhances the training process, leading to more refined and meaningful feature representations.

- The ablation studies presented in Tables 6 and 7 are thorough, covering various aspects of the proposed method. These studies systematically analyze the contribution of each component, reinforcing the effectiveness and necessity of the design choices.

- The method demonstrates a substantial performance improvement over existing baselines across multiple benchmarks. The quantitative results highlight its superiority in terms of both accuracy and robustness, confirming its effectiveness in handling divergent cases.

- The role of the ranking loss is comprehensively discussed, with detailed insights into how it influences the learning dynamics. The paper provides both empirical evidence and theoretical analysis showing how ranking-based supervision enhances model generalization and optimization.

- The proposed method also proves to be beneficial for distilling vision transformers, effectively transferring knowledge while preserving critical ranking information. This indicates its potential for improving the efficiency and performance of large-scale transformer-based architectures.

Weaknesses:

- One closely related work [1] incorporates intrinsic ranking information of logits using Pearson Correlation. Since both methods aim to enhance knowledge distillation by capturing ranking relationships, it would be valuable to compare the proposed ranking loss with this approach as an auxiliary loss with more discussions. This would provide deeper insights into their relative effectiveness and compatibility.

- A discussion on the numerical range of the ranking loss would be helpful, as it would provide better interpretability and understanding of its impact during training. Clarifying whether its scale remains stable across different datasets and architectures could further enhance reproducibility.

- The influence of the steepness parameter in Eq. 6 is not explicitly analyzed. Since this parameter likely affects the sensitivity of the ranking loss, a more detailed discussion would be beneficial to determine its optimal setting and impact on training stability and performance.

[1] Knowledge distillation from a stronger teacher, NeurIPS 2022.

**Questions For Authors:**

Please see weaknesses.

**Relation To Broader Scientific Literature:**

NA

**Theoretical Claims:**

Yes, theoretical claim is proved properly.

---

> ### Author Rebuttal · Authors · 2025-04-01
>
> We sincerely thank you for the in-depth feedback! We highly value each of your comments, and all your concerns are addressed point by point:
>
> ---
>
> **Q1. Discuss More techniques of detection improvement:**
>
> - We sincerely appreciate your suggestion. Exploring the application of the proposed method in downstream tasks is indeed one of our future directions. However, our method is primarily designed to enhance knowledge distillation. We have compared our approach with other knowledge distillation-based object detection methods, and the results are presented in Table 5 of the paper and in Table T2.2 of our response to Reviewer vqaG. In future versions, we will include discussions and potential integrations with pure object detection methods.
>
> ---
>
> **Q2. Comparison with Pearson Correlation:**
>
> - Thank you for your interest in our work. We also note the application of the Pearson correlation coefficient in distillation; however, unlike our proposed method, the Pearson correlation coefficient measures the linear relationship between two variables. Our method focuses on the ranking relationship of logits, which is not strictly a linear relationship. Therefore, the Kendall correlation coefficient may be more suitable for capturing this non-linear ranking information.
> - We have already compared the performance of DIST and our proposed method in the distillation tasks in **Table 1 and 2 of the paper**. To further investigate the combined effect of using both linear and non-linear auxiliary functions, we incorporated a ranking loss into DIST, which improved its performance. The results are shown in **Table T1.1**.
>
>   *Table T1.1: Experiments on Combining DIST.*
>   | Teacher -> Student | ResNet32×4 -> WRN-16-2 | ResNet32×4 -> WRN-40-2 | WRN-40-2 -> SHN-V1 | VGG13 -> VGG8 | WRN-40-2 -> WRN-40-1 |
>   | :---: | :---: | :---: | :---: | :---: | :---: |
>   | DIST[1]      | 75.58 | 78.02 | 76.00 | 73.80 | 74.73 |
>   | DIST[1]+Ours | 75.85(+0.27) | 78.54(+0.52) | 76.23(+0.23) | 74.06(+0.26) | 74.86(+0.13) |
>
> [1] Knowledge distillation from a stronger teacher, NeurIPS 2022
>
> ---
>
> **Q3. Range of the Ranking Loss:**
>
> - According to **Equations 5 and 6 in the paper**, the proposed ranking loss has a value range of [-1, 1]. We set the weight of ranking loss to be the same as that of KL divergence. For baselines using different KL divergence weights, the value of the ranking loss may vary, but its scale generally remains within the range of [-1, 1]. Across all teacher-student combinations with the same baseline and dataset, the ranking loss maintains consistent weight and scale.
>
> ---
>
> **Q4. Analysis of the Steepness Parameter:**
>
> - In Table 7 of our paper, we investigate the impact of the steepness parameter. We observe that although the optimal steepness parameter may vary across different teacher-student combinations, larger steepness parameters generally yield better results. This finding aligns with intuition: as the steepness parameter increases, the tanh function approximates the sign function more closely, making it more sensitive to ranking and less sensitive to scale differences.
>
> **Q5. Other Comments and Suggestions:**
>
> - Thank you very much for your feedback regarding the formatting of our paper. We will adjust these formulas in the future version of the manuscript. We appreciate your attention to detail.

---

### Decision · Program_Chairs · 2025-05-01

**Decision:**

Accept (poster)

**Comment:**

Standard knowledge distillation often uses KL divergence to align student-teacher logits. This paper argues this can lead to suboptimal solutions and neglect low-probability channels, hindering learning and inter-class relationship transfer. It then introduces an auxiliary ranking loss, based on Kendall's tau, to mitigate these issues. By constraining logit rank similarity, the proposed loss provides scale-invariant gradients, better captures inter-class information, and helps avoid KL divergence's pitfalls without altering its main objective. Theoretical analysis and experiments across multiple datasets/architectures suggest that this ranking loss can enhance various logit-based distillation methods.

Reviewers generally acknowledged the innovative nature of leveraging ranking information in this context and recognized the potential benefits. The reviewers were also generally positive about baseline experiments across diverse datasets using both CNN and ViT architectures, ablation studies, and helpful visualizations. However, some concerns were raised around the paper's positioning within the current literature and the rigor of its claims. Reviewers (Za63, vqaG, wbp7) raised concern about missing comprehensive comparisons with SOTA distillation methods, specifically methods focusing on inter-class relationships like WKD and DIST, as well as strong KL-based techniques such as NKD, CTKD, WTTM, and LSKD. Additionally, reviewers (wbp7, Zvoo) expressed skepticism about whether KL divergence truly "neglects" low-probability channels detrimentally or if the provided examples were sufficiently representative. Less critical concerns were the missing analysis of computational/memory overhead (wbp7), clarification on numerical range of the loss and hyperparameter effects (Za63), the choice of Kendall's tau over other ranking metrics (vqaG, wbp7), and the theoretical justification for certain claims (wbp7, vqaG).

In the rebuttal, the authors presented results demonstrating that their auxiliary ranking loss consistently improved the performance of several requested SOTA methods (including NKD, CTKD, WTTM, LSKD, DIST, and WKD) when used as a plug-in. They also showed the additional computational time was minimal. They further addressed technical questions regarding the loss range, the choice of Kendall's tau, and provided clarifications on derivations and theoretical points. This thorough response successfully resolved most major issues raised by (Za63, Zvoo, wbp7) and they increased their initial score. (vqaG) acknowledged the new results but maintained a weak accept, primarily due to a perceived lack of deep methodological discussion comparing ranking loss and WKD, despite the authors providing performance comparisons. Overall, the rebuttal significantly strengthened the paper's standing, leading to a final consensus leaning towards acceptance. In agreement with the reviewers, I find the idea of using Kendall's tau for knowledge distillation a promising direction, and recommend accept.